# The Interplay of Inflammation and Gut-Microbiota Dysbiosis in Alzheimer’s Disease: Mechanisms and Therapeutic Potential

**DOI:** 10.3390/ijms26188905

**Published:** 2025-09-12

**Authors:** Hanis Nabilah Abdol Samat, Nurul Nadirah Razali, Hazlina Mahadzir, Tengku Sifzizul Tengku Muhammad, King-Hwa Ling, Nur Izzati Mansor, Shahidee Zainal Abidin

**Affiliations:** 1Faculty of Science and Marine Environment, Universiti Malaysia Terengganu, Kuala Nerus 21030, Terengganu, Malaysia; hanisnabilah253@gmail.com (H.N.A.S.); nurulnadirahrazali@gmail.com (N.N.R.); 2Department of Medicine, Hospital Canselor Tuanku Muhriz, Faculty of Medicine, Universiti Kebangsaan Malaysia, Cheras 56000, Kuala Lumpur, Malaysia; drhazlina2013@gmail.com (H.M.); 3Institute of Climate Adaptation and Marine Biotechnology, Universiti Malaysia Terengganu, Kuala Nerus 21030, Terengganu, Malaysia; sifzizul@umt.edu.my (T.S.T.M.); 4Department of Biomedical Sciences, Faculty of Medicine and Health Sciences, Universiti Putra Malaysia, Serdang 43400, Selangor, Malaysia; lkh@upm.edu.my (K.-H.L.); 5Malaysian Research Institute on Ageing (MyAgeing^®^), Universiti Putra Malaysia, Serdang 43400, Selangor, Malaysia; 6Department of Nursing, Faculty of Medicine, Universiti Kebangsaan Malaysia, Cheras 56000, Kuala Lumpur, Malaysia; 7Research Interest Group Biological Security and Sustainability (BIOSES), Faculty of Science and Marine Environment, Universiti Malaysia Terengganu, Kuala Nerus 21030, Terengganu, Malaysia

**Keywords:** Alzheimer’s disease, gut microbiota dysbiosis, systemic inflammation, neuroinflammation, gut microbiota metabolite

## Abstract

Alzheimer’s disease (AD) represents a major global health challenge, characterised by progressive neurodegeneration that leads to cognitive decline. Inflammation is a key factor in the pathogenesis of AD, affecting both neuroinflammation and systemic inflammation. In AD, neuroinflammation is marked by the activation of microglia and the release of pro-inflammatory cytokines, which exacerbate neuronal damage and cognitive deficits. Systemic inflammation further compromises the blood–brain barrier (BBB), increasing its permeability and permitting the entry of inflammatory molecules and immune cells into the brain, thereby advancing the disease’s hallmark features. Recent studies have elucidated the influence of gut microbiota dysbiosis on AD and inflammation. This imbalance is thought to be associated with alterations in the concentrations of short-chain fatty acids (SCFAs) and bile acids, which can modulate neuroinflammation and contribute to AD pathology. Additionally, imbalances in neurotransmitters resulting from gut microbiota dysbiosis can further disrupt brain function and facilitate AD progression. This review provides an overview of the hypothesis that systemic and central nervous system (CNS) inflammation, together with gut-microbiota dysbiosis, may interact to influence the development and progression of AD.

## 1. Introduction

Alzheimer’s disease (AD) is the most common form of dementia, accounting for at least two-thirds of cases in individuals aged 65 and older [1]. Globally, more than 57 million people live with dementia, with AD representing the majority of cases [2]. The prevalence of AD increases sharply with age, affecting over 40% of individuals aged 85 and above [3]. Nearly 10 million new cases of dementia are reported each year, and this number is projected to reach 152 million by 2050 due to increased life expectancy and population ageing [4]. Women are disproportionately affected, accounting for nearly two-thirds of all cases, which is partially attributed to their longer life expectancy and biological factors [5]. AD is a multifaceted condition that manifests through a wide range of clinical symptoms, mainly cognitive but often accompanied by profound behavioural and functional impairments. The progression of AD varies between individuals, but it ultimately leads to complete dependency on caregivers and a reduction in the quality of life.

This disease is characterised by the early development of amyloid-β (Aβ) peptides in the brain, which form diffuse and neuritic plaques [6]. This is followed by neurofibrillary tangles (NFT) inside brain cells, which comprise hyperphosphorylated tau protein [7]. Despite multiple studies showing the presence of neurofibrillary tangles in the absence of Aβ deposition [8]. In addition, the disease involves further pathophysiological alterations and processes, including neuroinflammation, synaptic dysfunction, and metabolic dysregulation [9]. Although these biological pathways are pivotal in the AD pathology, mounting evidence suggests they are intricately linked to environmental factors. The incidence and progression of AD are increasingly recognised as being impacted not just by neuropathological changes but also by individual lifestyle choices, overlapping biological predispositions, and pre-existing health disorders.

Unhealthy dietary habits, especially those high in saturated fats and refined sugars, have been demonstrated to aggravate neurodegeneration and elevate the likelihood of acquiring AD [10]. Conversely, compliance with nutritious dietary regimens such as the Mediterranean diet (MeDi) or the ketogenic diet (KD), which are rich in antioxidants, vitamins, and omega-3 fatty acids, correlates with a reduced risk of cognitive decline and slower progression of AD [10]. These protective dietary approaches, combined with lifestyle factors such as regular physical exercise and smoking cessation, are believed to induce beneficial metabolic and molecular changes [11]. These changes may inhibit inflammation and oxidative stress, thereby potentially reducing the accumulation of amyloid plaques, NFT, and neuritic pathology in the brain.

Metabolic and vascular comorbidities (such as obesity, type 2 diabetes, hypertension, and cardiovascular disease) are recognised to exacerbate systemic inflammation, oxidative stress, and vascular dysfunction, all of which are key contributors to the pathogenesis of AD [11]. These effects are further amplified by genetic predispositions, particularly the presence of the APOE ε4 allele. While this allele is found in approximately 15% of the general population, it is present in up to 60% of AD patients [12]. Carriers of APOE ε4 face a markedly increased risk of developing AD, especially when exposed to additional modifiable risk factors, including chronic stress [12]. Even though extensive studies have been conducted to comprehend the pathophysiology of the disease, the cause-and-effect links of the many biological processes involved in AD remain incompletely understood.

The dysbiosis of gut microbiota has become more recognised for its role in the progression of neurodegenerative diseases, especially AD, by intensifying systemic and neuroinflammation. The gut microbiota colonises the human intestine from birth, establishing a bidirectional relationship with the host that evolves across life stages. Microbial diversity adapts to host lifestyle and diet, supplying essential metabolites (e.g., short-chain fatty acids, SCFAs) while receiving nutrients in return. In turn, the host modulates its metabolism and immune function in response to microbial shifts. This symbiotic interaction is critical for immune priming, as the microbiota shapes innate immunity through mechanisms such as Gram-negative bacterial membrane vesicle signalling [13]. Dysbiosis disrupts this equilibrium and is increasingly implicated in neurodegenerative diseases like AD, where it exacerbates systemic and neuroinflammation. Altered microbiota reduces SCFA production and elevates pro-inflammatory mediators that traverse the gut–brain axis, impairing microglial function [14]. Diet plays a pivotal role: refined sugars deplete beneficial taxa (e.g., *Bifidobacterium*, *Lactobacillus*) and elevate cortisol [15,16], whereas the MeDi enriches SCFA-producing genera (*Faecalibacterium*, *Roseburia*) and reduces pathogens (e.g., *Proteobacteria*) [17]. The changes in microbiota composition and metabolite production caused by diet may affect inflammatory pathways and microglial activation, highlighting the significance of dietary patterns as adjustable risk factors in the prevention and progression of many relevant diseases, including neurodegenerative ones, such as dementia and AD.

This review aims to summarise the main current understanding of gut microbiota dysbiosis and related metabolism with respect to pathways of central and systemic inflammatory response. AD exhibits significant genotypic and phenotypic heterogeneity, compounded by complex interactions with environmental factors (e.g., diet and microbiota composition), which influence disease onset, progression, and therapeutic response. Growing evidence implicates inflammation as a central mechanism in AD pathophysiology, serving not only as a consequence of Aβ and tau pathology but also as an active driver of neurodegeneration and cognitive impairment. Furthermore, this review examines the interplay between gut microbiota, inflammatory pathways, and core AD hallmarks. By synthesising key mechanisms, identifying therapeutic targets, and addressing knowledge gaps, we aim to advance the understanding of the gut–brain axis as a modifiable target for AD prevention and treatment.

## 2. Alzheimer’s Disease

AD is defined by a wide range of characteristics, encompassing both genotype and phenotype, along with complex interactions with environmental factors. The definition of the disease, rooted in histopathological or clinical-pathological criteria, may also lead to heterogeneity. Non-AD, including cerebrovascular disease and systemic diseases, also play a role in the clinical presentation and prognosis of AD. Even when characterised by pathological criteria, AD exhibits significant heterogeneity among subtypes, each with distinct demographic and clinical features.

### 2.1. Main Clinical, Anatomical and Histological Features

Cognitive impairment and dementia in older adults often result from a combination of brain pathologies, including neurodegenerative changes such as Aβ plaques, tau NFTs, α-synuclein, TAR-DNA-binding protein 43 (TDP-43), and various cerebrovascular lesions [18,19,20]. Notably, only about half of individuals diagnosed with AD during life exhibit purely Alzheimer’s-related pathology upon autopsy, without evidence of other coexisting brain diseases [21]. Conversely, many individuals diagnosed with non-AD dementias are later found to have AD pathology, indicating that mixed or overlapping brain pathologies are common and complicate accurate diagnosis. Furthermore, numerous elderly individuals who appeared cognitively normal near the time of death have been found to harbour AD-related brain changes at autopsy, such as Aβ plaques and tau tangles [22].

In a longitudinal study, many non-demented individuals showed pathological features typically associated with dementia [23]. In this study, the majority had amyloid deposits, all exhibited tau tangles, approximately 25% had macroscopic infarcts, another 25% had microinfarcts, and 5–10% had neocortical Lewy bodies. The presence of abnormal Aβ, neocortical Lewy bodies, and NFTs in these individuals was associated with subtle cognitive decline, even though they were not classified as demented. It remains uncertain whether such individuals would have developed clinical symptoms had they lived longer, but these findings highlight that the biological processes underlying dementia can begin long before cognitive symptoms become apparent.

Furthermore, individuals in the oldest age group are significantly more prone than their younger counterparts to experience multiple health issues linked to cognitive impairment. The estimated likelihood of dementia, along with its severity, was observed to rise with an increasing number of pathological diagnoses [24]. Individuals exhibiting intermediate to high severity of AD pathology were four times more likely to develop dementia in the presence of an additional non-AD pathology, suggesting that the impact of multiple pathologies could be either additive or synergistic. Karanth et al. [25] conducted a study involving 1346 participants, who were enrolled in two extensive cohorts, with longitudinal evaluation persisting until the time of autopsy. The study revealed diverse topographic patterns of brain atrophy, along with the common presence of additional pathologies and clinical phenotypes throughout life. The presence of α-synucleinopathy, specifically, was linked to more significant executive and visuospatial dysfunction.

Additional studies have indicated that the conformational variability of Aβ42 might be linked to the differences in the rates of progression of AD. Cohen et al. [26] explored the connection between various structural assemblies of Aβ and the rates of clinical decline in 48 cases of AD, each exhibiting notably different disease durations. Condello et al. [27] examined the structural variability of Aβ deposits present in both familial and sporadic AD, in addition to cerebral amyloid angiopathy. The pathological heterogeneity seen in AD includes differences in the distribution of NFTs. Murray et al. [28] identified 889 cases of AD from a brain bank database, all exhibiting a Braak NFTs stage greater than IV. Patients with hippocampal-sparing AD were younger at the time of death, experienced a shorter duration of the disease, and were more often male. In contrast, those with limbic-predominant AD were older at death, exhibited slower progression rates, and were more frequently female.

### 2.2. Demographic Features

The prevalence of AD varies according to demographic factors such as age, gender, population and ethnicity. Age remains the most significant risk factor for sporadic AD, affecting approximately 3% of individuals aged 65–74, 17% of those aged 75–84, and 32% of people aged 85 and older [29]. Sex-based differences in the risk and prevalence of AD and related dementias (ADRDs) are well established, with nearly two-thirds of diagnosed cases occurring in women [30]. This higher prevalence in women is largely attributed to their longer life expectancy, given that advanced age is the primary risk factor for AD. However, other contributing factors include social, health, and biological influences (such as the role of sex hormones in brain ageing and disease susceptibility) [31]. Despite extensive study, findings on age-specific gender differences in AD risk remain inconsistent, likely due to varying risk factor distributions across age groups, populations, and geographic regions.

Ethnic disparities in the risk of developing AD and related dementias are well documented. Elderly African-Americans face approximately double the risk compared to elderly people of European origin, while elderly individuals from Latin America have about 1.5 times the risk [32]. The Latin American population is notably diverse in terms of cultural background, genetic ancestry, and health status, and prevalence rates may vary across different subgroups [33]. Genetic factors alone do not fully account for these patterns. Instead, health conditions such as cardiovascular disease and type 2 diabetes (which are more common in these populations and are known to increase dementia risk) may play a central role [34]. In addition, social and economic factors, including lower educational attainment, higher poverty rates, and greater exposure to lifelong stressors and discrimination, are believed to contribute to the increased likelihood of developing AD [35].

### 2.3. Heterogeneity of Alzheimer’s Disease

Episodic memory is the primary and most characteristic cognitive domain affected in individuals with AD. However, a subset of patients may initially present with impairments in non-memory-related cognitive functions [5]. Classifying patients based on different cognitive patterns can offer useful prognostic information. This is especially valuable in treatment trials, where progression rates may vary across subgroups. The prodromal phase of dementia, known as mild cognitive impairment (MCI), is clinically and pathologically heterogeneous, with differing probabilities of progression to dementia [36]. MCI is typically categorised into two main subtypes: (1) amnestic MCI (aMCI), which is marked by prominent memory loss and is often a precursor to AD, and (2) non-amnestic MCI (naMCI), which involves impairments in other cognitive domains such as language, executive function, visuospatial skills, or behaviour. Individuals with naMCI are more likely to progress to non-Alzheimer’s dementias, including vascular dementia and dementia with Lewy bodies (DLB) [37].

Moreover, subgroups of people diagnosed with AD have been identified using various statistical methods, including factor and cluster analysis. Phillips et al. [38] examined cognitive data from 146 participants with early-onset AD (EOAD). Their cluster analyses revealed a four-cluster solution: (1) memory-dominant impairment, characterized by atrophy and hypometabolism in the medial and lateral temporal, lateral parietal, and posterior cingulate regions; (2) memory and visuospatial-dominant impairment, featuring atrophy and hypometabolism in the medial temporal, temporoparietal, and frontal cortices; (3) impairment in memory, language, and executive function, with atrophy in all regions except the sensorimotor cortex; and (4) global cognitive impairment, with widespread atrophy and hypometabolism throughout the brain. Based on the analysis, individuals with AD or related conditions may exhibit distinct patterns of cognitive decline, particularly in the visuospatial and language/executive domains. A longitudinal analysis using neuropsychological data from individuals with mild to severe AD, based on the U.S. National Alzheimer’s Coordinating Center (NACC) database, revealed that approximately 80% of patients with autopsy-confirmed AD exhibited a typical cognitive profile [39]. In contrast, atypical cognitive profiles were more commonly observed in younger age, males, those without the APOE ε4 genotype, and individuals with less severe dementia, higher depression scores, lower Braak stages at autopsy, and a more gradual cognitive decline.

Lam et al. [40] further identified distinct cognitive phenotypes linked to specific regional brain changes: (1) a pure amnestic syndrome with a slow decline and pathology confined mainly to the medial temporal lobe; (2) predominant language impairment associated with early onset, rapid decline, and left parietal atrophy or hypometabolism; (3) predominant visuospatial deficits related to right parietal lobe atrophy/hypometabolism; and (4) executive dysfunction disproportionate to memory loss, sometimes accompanied by behavioural symptoms such as disinhibition or apathy, and associated with frontal lobe atrophy. Similarly, Peter et al. [41], through principal components and cluster analysis of 24 cognitive scores, identified distinct cognitive subtypes and found that those with focal semantic impairments experienced a significantly more rapid progression than individuals in other cognitive clusters. Together, these findings underscore the heterogeneity in AD presentation and progression, highlighting the importance of personalised approaches to diagnosis and treatment.

## 3. Alzheimer’s Disease and Inflammation

The heterogeneity of AD is reflected in its various clinical presentations, neuropathological features, and progression patterns, and is closely linked to the complexity of neuroinflammation. Long-term inflammation plays a central role in the development and progression of AD. The accumulation of Aβ plaques and tau tangles activates immune cells like microglia, leading to the release of pro-inflammatory cytokines such as interleukin-6 (IL-6) and interleukin-1β (IL-1β). This chronic neuroinflammatory response damages neurons and accelerates cognitive decline. In addition, systemic inflammation, often indicated by elevated biomarkers like C-reactive protein (CRP) in AD patients, may further exacerbate neuroinflammation and contribute to disease progression.

### 3.1. Alzheimer’s Disease and Neuroinflammation

Neuroinflammation has been recognised as an additional AD characteristic. In 2019, a meta-analysis study summarised studies investigating the variation in neuroinflammation using positron emission tomography (PET) imaging with translocator protein (TSPO) ligands. TSPO is a mitochondrial outer membrane protein highly expressed in activated microglia and astrocytes during neuroinflammation. This ligand is involved in critical mitochondrial functions, including energy production, cholesterol transport, steroidogenesis, and modulation of reactive oxygen species (ROS) generation. In AD, TSPO is upregulated in response to Aβ plaques and tau pathology, making it a target for PET imaging studies. Ligands such as [11C] PK11195 bind to TSPO, enabling in vivo visualisation of microglial activation, which correlates with disease progression and cognitive decline [42,43,44,45]. The analysis compared normal controls, individuals with MCI, and patients with AD. The conclusive findings indicated that the levels of neuroinflammation in patients with AD were elevated and more extensive compared to individuals with MCI, whose neuroinflammation levels were only moderate [42]. This increased neuroinflammation is closely associated with the presence of Aβ plaques and tau tangles, both of which exacerbate neuroinflammatory responses. Furthermore, multiple PET studies that focused on TSPO and amyloid or tau ligands discovered a strong association between the extent of neuroinflammation during the early stage of MCI and the levels of Aβ [43,44,45], highlighting the interplay between neuroinflammation and these key AD hallmarks.

In AD, neuroinflammation exacerbates cognitive impairment and the course of the disease by promoting synaptic dysfunction and neuronal damage. Tumour necrosis factor-alpha (TNF-α), IL-1β, and IL-6 are examples of pro-inflammatory cytokines that impede synaptic plasticity, alter synaptic transmission, and promote neuronal death, all of which can result in cognitive deficiencies [46]. These inflammatory cytokines contribute to the progression of AD pathology by enhancing the accumulation of Aβ plaques and accelerating tau tangle formation [47]. Additionally, nitric oxide (NO) and prostaglandins are two neurotoxic factors that contribute to the acceleration of neurodegenerative processes in AD. These substances are produced when chronic neuroinflammation persists [46].

### 3.2. Microglial Activation

Microglia are the primary immune cells in the CNS and are activated in response to injury or infection [48]. In normal conditions, the microglia monitor the brain, support the growth of new neurons, and facilitate the development of connections between neurons through the brain-derived neurotrophic factor (BDNF) signalling pathway to maintain the balance of the brain [49]. During immunological surveillance, microglia detect endogenous signals (known as danger-associated molecular patterns, DAMPs) and exogenous signals (known as pathogen-associated molecular patterns, PAMPs) using pattern recognition receptors (PRRs) such as scavenger receptors (SRs), Toll-like receptors (TLRs) and triggering receptor expressed on myeloid cells 2 (TREM2) [50]. These receptors allow microglia to recognise and bind to pathological proteins, initiating the activation cascade.

However, in AD, microglial activation is triggered by soluble Aβ oligomers, lipopolysaccharides (LPS), fibrils, and tau tangles that bind to the PRRs, leading to neuroinflammatory responses (Figure 1) [51]. Activated microglia engulf and internalise Aβ and tau aggregates through phagocytosis. This process is crucial for clearing pathological proteins and reducing their accumulation in the brain. The activation of microglia involves various intracellular signalling pathways, including nuclear factor kappa B (NF-κB), mitogen-activated protein kinases (MAPKs), and inflammasome activation [52]. These pathways regulate pro-inflammatory cytokines, such as IL-1β and TNF-α, in response to Aβ and tau tangle, contributing to neuroinflammation, neuronal damage, and synaptic dysfunction in AD. The excessively activated microglia also produce ROS and NO, further exacerbating oxidative stress and neurotoxicity [53]. Clinically measurable pro-inflammatory cytokines include IL-1β, TNF-α, and IL-6, which are often evaluated as biomarkers for inflammation in AD and other neurodegenerative conditions.

IL-6 is strongly implicated in the cognitive decline observed in AD. A study by Wennström et al. [54] reported significantly lower levels of IL-6 in cerebrospinal fluid (CSF) from patients with DLB compared to both AD patients and cognitively healthy controls. Furthermore, elevated plasma IL-6 has been inversely correlated with cognitive performance and with the volumes of the hippocampus and hypothalamus in a cohort of AD patients and cognitively healthy controls. In AD mouse models (APPswe/PS1ΔE9), neutralisation of IL-6 in the brain improved memory performance, corrected peripheral glucose intolerance and reduced circulating IL-6 levels [55]. Similar to other biomarkers, TNF-α and IL-1β have also been found at elevated levels in the APPswe/PS1ΔE9 AD mouse model relative to controls [56]. As neurodegeneration progresses, DAMPs, including chromogranin A and myeloid-related protein 14 (MRP14), can emerge and activate microglia [57,58]. These DAMPs may impair the microglia’s ability to clear Aβ effectively, thereby contributing to the persistent neuroinflammation characteristic of AD. Together, these inflammatory and DAMP-related changes highlight a complex inflammatory network contributing to AD pathogenesis.

### 3.3. Astrocytes

In addition to microglia, astrocytes are also key players in neuroinflammation and are closely associated with AD (Figure 2). Astrocytes, which are essential for maintaining homeostasis and supporting neuronal function in the CNS, undergo significant changes during neuroinflammation that can influence disease progression. In AD, astrocytes become reactive, releasing inflammatory molecules that can exacerbate neuronal damage and contribute to the formation of Aβ plaques [59].

In the early stages of AD, astrocytes play a protective role by attempting to clear extracellular Aβ through phagocytosis, thereby helping to reduce the accumulation of these plaques in the brain [60]. However, as AD progresses, astrocytes become increasingly impaired in this role, and their efficiency in clearing Aβ diminishes significantly. This functional decline not only reduces the brain’s ability to manage plaque buildup but also contributes to a shift in astrocytes’ behaviour, leading them to release a range of pro-inflammatory factors. The release of these molecules creates a vicious cycle in which inflammation accelerates plaque aggregation and further exacerbates neurodegeneration. For instance, IL-6 released by astrocytes in AD is closely associated with cognitive decline [55]. It is released in response to Aβ and oxidative stress, creating a toxic environment for neurons. The cytokine’s pro-inflammatory actions amplify the inflammatory response by recruiting and activating more immune cells. IL-6 also compromises blood–brain barrier (BBB) integrity, enabling peripheral immune cells to enter the brain, thereby increasing neuroinflammation [61].

Similarly, TNF-α plays a central role in the AD inflammatory response by interacting directly with neuronal receptors, triggering pathways that lead to cell death [62]. Elevated levels of TNF-α in the AD brain are linked to neuronal apoptosis and synaptic loss, both of which are central to the cognitive deficits seen in AD [63]. TNF-α disrupts synaptic signalling by interfering with glutamate regulation, causing excitotoxicity, a condition where excessive glutamate damages neurons [64]. Through these mechanisms, IL-6 and TNF-α contribute to a harmful inflammatory environment that accelerates AD progression, underscoring the central role of astrocyte-mediated neuroinflammation in driving the disease. Astrocytes release chemokines such as CCL2 (MCP-1) and CXCL10, which play a significant role in recruiting other immune cells to sites of plaque formation [65,66]. By attracting microglia and peripheral immune cells to areas where Aβ has accumulated, chemokines amplify the inflammatory response around plaques [67]. While the initial goal of this response is to contain and clear Aβ, the chronic nature of AD pathology leads to prolonged recruitment of immune cells, resulting in sustained inflammation that exacerbates neuronal damage. This influx of immune cells not only increases inflammation locally but also accelerates the cycle of neuroinflammation, as astrocytes and microglia continuously stimulate one another in the presence of Aβ.

Besides that, astrocytes also release complement proteins, particularly C1q and C3, which play a detrimental role in synaptic loss by tagging synapses for removal [68]. This tagging process is typically beneficial in immune responses, where it helps identify and clear pathogens. However, in the AD brain, the complement system becomes overactivated due to the chronic presence of Aβ plaques and the resulting inflammatory environment [69]. Activated astrocytes increase their production of complement proteins, which adhere to nearby synapses, marking them as targets for elimination. Microglia, the brain’s resident immune cells, have receptors that recognise these tagged synapses. Through the complement receptor, microglia bind to the C1q protein on these synapses and proceed to engulf and degrade them in a process called synaptic pruning [70]. While synaptic pruning is normal and necessary during brain development, in AD, it becomes excessive and maladaptive, leading to widespread and unintended synaptic loss. The continued presence of Aβ and inflammatory signals keeps astrocytes and microglia in an activated state, perpetuating the release of complement proteins and promoting further synaptic loss. This chronic cycle of inflammation and synapse removal erodes the neuronal network essential for cognitive functions like learning and memory. As synapses are lost, the brain’s communication networks degrade, leading to the progressive cognitive decline characteristic of AD. This mechanism, driven by astrocytes and microglia, underscores how neuroinflammation and complement protein activity can accelerate neurodegeneration in AD.

### 3.4. Alzheimer’s Disease and Systemic Inflammation

The brain is protected by tight barriers such as the BBB, CSF, and arachnoid barriers. These barriers provide a balanced microenvironment and protect against external insults like toxins, infectious agents, and peripheral pro-inflammatory cytokines. However, systemic inflammation could exacerbate neuroinflammation through the activation of immune cells within the brain, such as microglia and astrocytes [71]. These activated immune cells release pro-inflammatory excessively, including cytokines and chemokines, which can damage neurons and disrupt neural circuits. Systemic inflammation can increase BBB permeability, allowing inflammatory molecules, immune cells, and peripheral pathogens to enter the brain more easily [72]. Disruption of the BBB facilitates the infiltration of pro-inflammatory cytokines, such as IL-1β, IL-6, and TNF-α, into the brain parenchyma, contributing to neuroinflammation [73].

Several clinical studies have indicated a link between systemic inflammation and the subsequent onset of AD. Several studies have observed that certain proteins in plasma, such as IL-6, are highly associated with cognitive impairment and dementia risk [74,75,76]. Moreover, it was found that individuals with AD have high levels of TNF-α, IL-1β, and IL-6 production compared to non-demented elderly [77]. Alzheimer’s patients who have elevated levels of TNF-α in their blood and experience intermittent systemic infections experience a significant acceleration in cognitive decline [78]. Systemic inflammation is often accompanied by the production of ROS, which can damage cellular components and contribute to oxidative stress. ROS can directly damage endothelial cells and tight junction proteins, compromising the BBB [79]. A previous study has shown that systemic inflammation is closely linked to gut microbiota dysbiosis. Imbalances in the gut microbiota can lead to an increased production of pro-inflammatory cytokines and ROS, which perpetuate inflammation and oxidative stress. This dysbiosis contributes to the breakdown of the gut barrier, allowing microbial products such as LPS to enter the bloodstream, thereby exacerbating systemic inflammation and impacting the BBB [80].

The oxidative stress induced by systemic inflammation leads to lipid peroxidation, protein oxidation, and DNA damage in endothelial cells, subsequently exacerbating BBB permeability [80]. Inflammation also activates endothelial cells, making them more permeable. This activation results in the upregulation of adhesion molecules, such as intercellular adhesion molecule-1 (ICAM-1) and vascular cell adhesion molecule-1 (VCAM-1), which facilitate the transmigration of immune cells across the BBB [81]. Moreover, inflammatory signals can increase the expression and activation of matrix metalloproteinases (MMPs), particularly MMP-2 and MMP-9. These enzymes degrade the extracellular matrix and tight junction proteins, increasing the BBB permeability [82]. In APOE ε4 carriers, activation of the cyclophilin A-matrix metalloproteinase-9 (CypA–MMP9) pathway leads to disruption of BBB integrity, contributing to neuronal and synaptic dysfunction [83]. Additionally, NF-κB contributes to this process by transcriptionally upregulating MMP9 expression in cerebral vessels [84]. Notably, inhibition of this pathway in ApoE4 knock-in mice restored BBB integrity and improved neuronal and synaptic function [85]. Furthermore, genetic suppression of the CypA–MMP9 pathway in Apoe^−^/^−^ mice reversed neurodegenerative changes, and treatment with SB-3CT, an MMP9 inhibitor, eliminated MMP9 gelatinase activity and reversed BBB leakage [85].

The breakdown of the BBB in AD has been confirmed by over 20 independent postmortem human studies [83]. These studies revealed evidence of brain capillary leakage and perivascular accumulation of blood-derived components such as fibrinogen, thrombin, albumin, immunoglobulin G (IgG), and hemosiderin. Additionally, they reported degeneration of pericytes and endothelial cells, loss of tight junctions, and the migration of red blood cells (RBCs). While BBB weakening is a hallmark of AD, it can also occur independently with ageing and may be exacerbated by factors such as the APOE ε4 genotype, oxidative stress, inflammation, and the accumulation of Aβ or ROS [86]. Small artery dysfunction disrupts the BBB, reduces cerebral blood flow and amyloid clearance, and contributes to neuronal damage. The infiltration of immune cells into the CNS can further disrupt the BBB and promote neuroinflammation. In addition, a previous study has found that individuals with cognitive impairment showed higher levels of systemic inflammation and increased activation of microglia when compared to healthy individuals [87]. They activate microglia within the brain, releasing additional pro-inflammatory cytokines and chemokines [88]. This glial cell plays a crucial role in maintaining BBB integrity, and its activation can lead to the disruption of tight junctions and increased BBB permeability [89]. The interaction between activated glial cells and endothelial cells can amplify the inflammatory response, further compromising the BBB. Systemic inflammation can significantly increase BBB permeability by releasing cytokines, oxidative stress, endothelial activation, MMP activity, and glial cell activation.

## 4. Association of Gut-Microbiota and Alzheimer’s Disease

Gut microbiota diversity and composition have been widely studied in AD, revealing significant alterations compared to healthy controls (HCs). A meta-analysis study summarised, at the individual study level, findings on alpha diversity were inconsistent [90]. However, five studies reported a significantly lower Shannon index in AD spectrum patients, suggesting a reduction in microbial richness and evenness [91,92,93,94,95]. In contrast, beta diversity analyses were more consistent. Out of 11 studies, nine demonstrated clear clustering separation between AD patients and HCs, indicating substantial shifts in gut microbial communities [92,93,94,95,96,97,98,99,100]. These findings collectively highlight a disrupted microbial ecosystem in AD, although variability between cohorts remains a challenge in interpretation.

Firmicutes and Bacteroidetes were the predominant phyla in the gut microbiota, accounting for approximately 90% of the total microbial population [90]. Within Firmicutes, genera such as *Ruminococcus*, *Faecalibacterium*, *Lachnospira*, *Dialister*, *Lachnoclostridium*, and *Roseburia* were reduced in AD patients, while *Phascolarctobacterium* and *Lactobacillus* were increased. Regarding the Bacteroidetes phylum, the overall abundance of *Bacteroides* did not differ significantly between AD spectrum patients and HCs. However, differences were observed across ethno-racial populations [90]. Subgroup analyses showed increased levels of *Bacteroides* and *Alistipes* in American and Egyptian cohorts, whereas Chinese cohorts exhibited reduced levels of these genera. In contrast, *Prevotella* abundance showed no consistent pattern across populations. Within the Actinobacteria phylum, primarily represented by *Bifidobacterium*, pooled data showed no overall association with AD. Yet, a positive correlation between *Bifidobacterium* abundance and AD was specifically observed in Chinese cohorts. Meanwhile, *Akkermansia muciniphila*, the sole representative of the Verrucomicrobiota phylum, was consistently elevated in AD patients across multiple studies.

Further evidence linking gut dysbiosis to AD pathogenesis arises from studies examining inflammation-related gut microbiota. In patients with cerebral amyloidosis (Amy+), levels of *Eubacterium rectale* and *Bacteroides fragilis* were reduced, while *Escherichia/Shigella*, a genus commonly associated with inflammation, was significantly increased [101]. These microbiota alterations were accompanied by elevated levels of pro-inflammatory cytokines such as IL-6, CXCL2, NLRP3, and IL-1β, and a reduction in anti-inflammatory IL-10. A strong correlation was observed between the presence of *Escherichia/Shigella* and the increased levels of inflammatory markers, suggesting a role for gut-derived inflammation in the progression of AD. Supporting this inflammatory link, a study from the Kazakhstan population identified notable associations between specific bacterial taxa and serum biochemical markers [102]. For example, adiponectin was positively correlated with *Faecalibacterium*, Actinobacteria, and Christensenellaceae, while CRP levels were linked to Firmicutes, Lachnospiraceae, and *Klebsiella pneumoniae*. These findings further highlight the interaction between gut microbial alterations and systemic inflammation, reinforcing the hypothesis that gut dysbiosis contributes to neuroinflammatory processes in AD.

Beyond bacteria, gut fungi may also play a role in early neurodegenerative changes. A study examining gut mycobiota in patients with aMCI found increased levels of fungal families such as Sclerotiniaceae, Phaffomycetaceae, Trichocomaceae, and Cystofilobasidiaceae, and genera such as *Kazachstania* and *Cladosporium*. *Meyerozyma*, by contrast, was reduced [103]. These alterations support the growing hypothesis that gut microbial dysbiosis contributes to AD progression. Collectively, these findings underscore the importance of the gut–brain axis in AD, with gut microbiota diversity, composition, and inflammatory interactions serving as potential biomarkers and therapeutic targets.

## 5. Gut-Microbiota Dysbiosis, Inflammation and Alzheimer’s Disease

Gut microbiota dysbiosis is strongly associated with systemic inflammation. Dysbiosis disrupts gut barrier integrity, allowing bacterial metabolites and endotoxins like LPS to enter the bloodstream. This triggers systemic inflammation by activating immune responses and releasing pro-inflammatory cytokines such as IL-6, IL-1β, and TNF-α. Chronic systemic inflammation, in turn, affects the brain, contributing to neuroinflammation. This gut–brain communication, mediated through the vagus nerve, microbial metabolites (like SCFAs), and immune signalling, links gut dysbiosis to inflammation-driven pathologies, including AD (Figure 3).

### 5.1. Lipopolysaccharide

The LPS is a constituent of the outer membrane of Gram-negative bacteria that possesses pro-inflammatory characteristics. It has been recognised as a significant element that contributes to the start and advancement of systemic inflammation [104]. LPS adversely affects gut function, which could cause intestinal inflammation and disrupt the organisation of tight junctions [105]. The LPS regulates the pro-inflammatory response that occurs through the involvement of TLR4 and cluster of differentiation 14 (CD14). TLR4 is a member of the PRR family and is found in several immune cells, including monocytes, macrophages, and Kupffer cells.

The recognition of LPS by TLR4 is facilitated by the presence of LPS-binding protein and CD14 [106]. Activation of TLR4 initiates two signalling pathways. The first pathway involves TIRAP and MyD88 adaptor proteins and begins at the plasma membrane. The second pathway, dependent on TRAM and TRIF, is activated in early endosomes when the receptor is internalised through endocytosis, regulated by a GPI-anchored protein called CD14. Thus, LPS-induced systemic inflammation is influenced by the rate at which TLR4 endocytosis is transported via the endo-lysosomal compartment. In addition, the activation of the MyD88- and TRIF-dependent signalling pathways occurs by the LPS-activated TLR4 from the plasma membrane to the endosomes, which leads to lysosome degradation and an inflammatory response.

A previous study used rats as a model for endotoxemia-induced sepsis to ascertain the relationship between neuroinflammation, soluble Aβ generation, Aβ plaque formation, and hyperphosphorylated tau deposition in the brain [107]. In this study, rats received a single intraperitoneal injection of LPS at a dose of 10 mg/kg. Findings showed that LPS endotoxemia led to elevated levels of cytokines (IL-1β, TNF-α, and IL-6) in both the blood and brain. Additionally, LPS treatment increased soluble Aβ and p-tau levels across the brain and resulted in a higher microglial density in LPS-treated rats compared to controls. Furthermore, administering repeated systemic LPS injections results in a protracted increase in A levels and cognitive impairments [108,109]. Remarkably, pregnant mice that were repeatedly exposed to systemic LPS develop AD-related characteristics, such as behavioural and neuropathological abnormalities, in their offspring [110]. Studies on Tg2576, PDAPP, APPSwe, APP/PS1, and App^NL-G-F^ mice demonstrate that there is an elevation in the accumulation of Aβ deposits when these mice are exposed to LPS-induced systemic inflammation [111,112,113,114]. Moreover, the administration of polyinosinic-polycytidylic acid (poly I:C) to 4-month-old 3xTg-AD mice results in the development of systemic inflammation, which subsequently leads to an increase in Aβ deposition in the brain [115].

### 5.2. Short-Chain Fatty Acid

SCFAs, such as acetate, propionate, and butyrate, are byproducts generated during the bacterial breakdown of dietary fibre [116]. These substances possess anti-inflammatory characteristics and aid in maintaining the integrity of the intestinal barrier [117]. Gut microbiota dysbiosis can result in reduced levels of SCFAs, which can cause an increase in the permeability of the gut barrier. This increased permeability allows pro-inflammatory chemicals to enter the bloodstream and initiate systemic inflammation. A previous study has demonstrated a reduction of six bacterial species that produce butyrate in subjective cognitive decline patients [118]. These include *Anaerostipes*, *Roseburia*, *Lachnospiraceae* UCG-004, *Ruminococcaceae* UCG-013, and the *Ruminococcus gnavus* group.

The results of these experiments were consistent with other studies that found a correlation between gut dysbiosis and decreased levels of *Lachnospiraceae*, *Ruminococcaceae*, or *Anaerostipes* in AD patients [96,119]. In addition, a decrease in the abundance of bacteria that produce butyrate, such as species belonging to the *Eubacterium*, *Clostridium*, and *Butyrivibrio* genera, was found to be linked to the disruption of the anti-inflammatory P-glycoprotein pathway in individuals with AD [97]. In addition, a decrease in the production of butyrate by the bacterium *Faecalibaculum* resulted in increased permeability of the gut and the BBB, which led to the development of neuroinflammation and the progression of cognitive impairment in mice due to the presence of microbial LPS [120].

Several studies have highlighted the beneficial effects of butyrate on cognitive function and neuropathology in mouse models. In one study, administration of butyrate-producing bacteria (*Clostridium butyricum*) directly into the stomachs of APP/PS1 mice mitigated cognitive decline, reduced Aβ plaque accumulation, and decreased microglial activation, along with lower production of pro-inflammatory cytokines (IL-1β, TNF-α, and IL-6) in the brain [121]. Additionally, the study demonstrated that butyrate treatment reduced CD11b and COX-2 levels and suppressed NF-κB p65 phosphorylation in Aβ-stimulated BV2 microglia. These findings suggest that *Clostridium butyricum* treatment may attenuate microglia-mediated neuroinflammation through modulation of the metabolite butyrate. Another study used the histone deacetylase (HDAC) inhibitor sodium butyrate (NaB) to mitigate memory deficits in the 5xFAD mouse model of AD [122]. The results indicated that NaB treatment significantly reduced Aβ levels by 40% and improved learning and cognitive performance in the 5xFAD mice. These findings indicate that butyrate could be used as a therapy to prevent and slow down the development of AD.

### 5.3. Microbial-Derived Amyloid

Microbial-derived amyloids produced by gut microbiota have been increasingly recognised for their role in gut microbiota dysbiosis and systemic inflammation. These amyloids, primarily produced by bacteria such as *Escherichia coli* and *Bacillus subtilis*, can influence host physiology and immune responses as they are biochemically similar to disease-associated amyloid structures [123]. Microbial amyloids can exacerbate inflammatory processes by activating immune receptors like TLR2, producing pro-inflammatory cytokines [124]. This can contribute to a chronic inflammatory state, often seen in conditions associated with gut dysbiosis, such as inflammatory bowel disease (IBD) and metabolic disorders.

A previous study demonstrated an elevated level of bacterial amyloid-curli in the large intestine of Tg2576 AD mice during the pre-symptomatic stage before developing Aβ pathology in the brain [125]. The presence of a significant amount of bacterial amyloid curli in Tg2576 mice was found to be associated with the loss of gut barrier integrity, which allows the translocation of amyloid-curli and the activation of TLR2 in aged AD mice [126]. In addition, the gut lumen contains dysbiotic pathogenic bacteria that can build biofilms and have the potential to develop amyloid-like proteins [127,128,129]. For instance, *Salmonella* spp. able to produce amyloid curli fibrils, specifically stimulating TLR2 signalling in animal models [130]. This stimulation also leads to an increase in Nos2 expression.

Bacterial amyloids can stimulate TLR2/9 receptors, contribute to memory impairments, and exacerbate white matter damage in AD [129,130,131]. Several studies have demonstrated that activating TLR2 leads to an increase in the aggregation of amyloid proteins, whereas inhibiting TLR2 reduces the levels of neuronal Aβ in the AD mouse model [132,133]. Based on these findings, the activation of TLR2 from the gut to the brain in response to bacterial curli may significantly impact the central Aβ pathology of AD.

### 5.4. Bile Acids

Bile acids are critical signalling molecules derived from cholesterol that play a central role in regulating metabolic and immunological processes. The synthesis of primary bile acids occurs in the liver cell through cholesterol oxidation. Oxysterols, oxidised cholesterol metabolites, stimulate nuclear receptors glycocholic acid (GCA) associated with neurodegenerative diseases [134,135]. A previous study noted a pattern of elevated levels of GCA, glycodeoxycholic acid (GDCA), and glycochenodeoxycholic acid (GCDCA) in aMCI and AD, even though these findings are not statistically significant [136]. Furthermore, a separate study found that patients with aMCI and AD have notably increased amounts of deoxycholic acid (DCA), lithocholic acid (LCA), and GDCA acids in their blood plasma [137]. In addition, there have been reports of substantial elevations in the plasma levels of the secondary bile acid glycoursodeoxycholic acid (GUDCA) in AD [138].

Another study profiling 22 bile acids in the brain and plasma of AD patients and APP/PS1 mice revealed associations between specific bile acids and AD pathology [139]. The results showed that cholic acid (CA) levels were significantly lower in the plasma of AD patients compared to age-matched controls. In contrast, CA levels were elevated in APP/PS1 mice, while hyodeoxycholic acid was reduced. In the AD brain, taurocholic acid (TCA) was significantly lower than in control subjects. While in APP/PS1 mice demonstrated higher brain levels of LCA and reduced tauromuricholic acid (TMCA), along with alterations in five other bile acids, such as CA, β-muricholic acid, Ω-muricholic acid, TCA, and tauroursodeoxycholic acid. These findings suggest that bile acid levels are notably dysregulated in AD.

Moreover, bile acids can modulate inflammation by interacting with various receptors, such as the farnesoid X receptor (FXR) and G protein-coupled bile acid receptor 1 (TGR5). Gut microbiota dysbiosis-induced alterations in bile acid metabolism impair FXR signalling, leading to increased intestinal permeability and inflammation [140]. Additionally, certain bile acids, like LCA, can directly activate pro-inflammatory pathways by engaging TGR5 on immune cells, promoting the release of cytokines and chemokines that exacerbate inflammation. Gut microbiota dysbiosis could also increase the proportion of these pro-inflammatory bile acids, thereby contributing to systemic inflammation and metabolic dysfunctions such as non-alcoholic fatty liver disease and type 2 diabetes [141].

### 5.5. GABA Neurotransmitter

The gut microbiota in the gastrointestinal tract enhances the host’s physiological function by synthesising metabolites and neurotransmitters and facilitating nutritional digestion [142,143]. Alterations in the gut microbial populations can be attributed directly to shifts in neurochemical equilibrium, increasing the physiological need for microorganisms to produce and control neurotransmitters as sources of energy [144,145]. Prominent gut microbial taxa, especially those altered in AD, such as *Bacteroides* and *Lactobacillus*, are known for their ability to synthesise the inhibitory neurotransmitter, γ-aminobutyric acid (GABA), through enzymes like glutamate decarboxylase [146,147]. Although the exact role of GABA in the gut of AD patients remains unclear, it has shown potential effects on mucin expression, immune regulation, and intestinal integrity.

The function of GABA in the gastrointestinal tract during AD may involve activating the vagus nerve to transmit sensory signals to the brain or modifying the integrity of the gut mucosal membrane barrier by increasing mucin synthesis [147,148,149]. The mucus layer in the gastrointestinal tract serves as a physiological barrier between microorganisms and the gut epithelium, enabling the transport of solutes into and out of the gut [147,150]. The reduction of mucus layers facilitates the closer proximity of microorganisms and their metabolites to the gut epithelial cells, compromising the integrity of tight junctions and enabling the entry of microbial products into the lamina propria. For this purpose, endogenous GABA produced by gut microbiota may interact with local neural and immune cells in the intestinal lamina propria. Instead of entering systemic circulation and crossing the blood–brain barrier, it is more likely to influence brain function indirectly through vagal signalling, immune modulation, or neuroendocrine pathways.

In AD, GABA is linked to decreased expression of claudin-5, one of the essential tight junction proteins in the brain, associated with cognitive decline. Knockout mice with reduced claudin-5 expression demonstrate impaired long-term potentiation due to enhanced GABAergic transmission, suggesting that GABA dysregulation might contribute to cognitive deficits in AD [148]. Furthermore, vagus nerve stimulation (VNS) has been shown to modulate neurotransmitter levels, including GABA, potentially improving neuroplasticity and behaviour in AD patients [151,152]. Although vagotomy increases GABA production in the gut [153], it remains unclear if VNS directly affects gut GABA levels. In the enteric nervous system, GABA regulates gastrointestinal motility and secretions through enteric neurons and glial cells and may also inhibit inflammation via the NF-κB pathway in glial cells [154]. While GABA’s role in gut health and neuroinflammation is recognised, its broader implications, particularly in diseases like AD, require further study to fully understand its complex and disease-specific effects.

### 5.6. Tryptophan

Tryptophan (Trp) is an essential amino acid obtained from dietary sources, crucial for protein production, development, and the maintenance of cellular functioning. Trp is a precursor for various biologically important compounds, such as the neurotransmitter serotonin and the hormone melatonin [155]. The majority of dietary tryptophan is metabolised through the kynurenine pathway, primarily by the enzymes indoleamine-2,3-dioxygenase 1 (IDO1) and tryptophan 2,3-dioxygenase (TDO), resulting in the conversion to NAD^+^ for energy or to kynurenine derivatives, including kynurenic acid (KYNA) and quinolinic acid (QUIN).

Gut bacteria metabolise Trp into indole and its derivatives [155]. These compounds may function as ligands for the aryl hydrocarbon receptor (AhR), a transcription factor present in both the intestine and central nervous system (CNS). AhR modulates intestinal homeostasis, immune responses, and neuroinflammation. Endogenous AhR ligands, including kynurenine and KYNA, influence AHR’s function in immune control. Activation of AhR in glial cells may elicit either proinflammatory or anti-inflammatory responses, contingent upon the presence of particular ligands. In the context of LPS-induced inflammation, the lack of AhR ligands leads to increased neurotoxicity, while their presence mitigates microglial proinflammatory activity by restricting iNOS and TNF-α production.

Trp metabolism additionally affects the interaction between microglia and astrocytes in the CNS [156]. Trp derivatives, including indole-3-propionic acid (IPA) and indole-3-aldehyde (I3A), can activate microglia, which subsequently communicate with astrocytes through AhR-mediated pathways [157]. Dietary Trp additionally stimulate microglia to secrete factors such as TGF-α and VEGF-β, hence modifying astrocyte function. Given that both neurones and glial cells produce AhR, these interactions profoundly influence neuroimmune signalling and CNS homeostasis. A diet lacking in Trp alters gut microbiota composition, elevates systemic inflammation, and affects the microbiota–gut–brain axis, a bidirectional communication network linking the central and enteric neural systems [158]. This axis encompasses neurotransmitter signalling, including serotonin and KYNA, which are vital for both cerebral and gut function. Decreases in these metabolites are associated with gut dysbiosis, impaired intestinal barrier integrity, and possibly, AD aetiology.

## 6. Potential Therapeutic Approaches Targeting the Gut–Brain-Axis in Alzheimer’s Disease

Many studies have shown that gut dysbiosis could be a contributing factor in the development and progression of AD. Gut dysbiosis triggers several immune and inflammatory pathways that exacerbate both systemic and neuroinflammation, contributing to disease progression. Key inflammatory pathways include NF-κB, NLRP3 inflammasome, and TLRs. When gut dysbiosis leads to an increase in LPS, these endotoxins bind to TLR4 receptors, activating NF-κB and the NLRP3 inflammasome. This results in the production of pro-inflammatory cytokines such as IL-1β, IL-6, and TNF-α, which circulate and trigger neuroinflammation upon crossing the BBB. Gut–brain communication is also mediated by the vagus nerve, which is influenced by microbial metabolites and neuroactive compounds like SCFAs and GABA.

Furthermore, systemic inflammation often compromises BBB integrity, as elevated cytokines degrade tight junction proteins (e.g., claudin-5) that maintain BBB structure. This degradation increases BBB permeability, allowing inflammatory molecules and microbial metabolites to enter the brain, intensifying neuroinflammation, tau pathology, and neuronal damage, all key features of AD pathology. Given the profound influence of the gut–brain axis on CNS health, there is growing interest in therapeutic approaches targeting this axis as a potential intervention for AD. Modulating gut health through probiotics, prebiotics, faecal microbiota transplantation (FMT), and dietary interventions may help restore microbial balance, reduce inflammation, and protect neuronal function (Table 1). These approaches aim to address AD at its systemic and molecular roots by enhancing gut health, supporting the BBB, and reducing neuroinflammatory processes that drive cognitive decline in AD.

### 6.1. Microbiota-Based Therapies

Microbiota-based therapies, including probiotics, have been proposed as a potential supplementary therapy for AD due to their influence on the gut microbiota, which may, in turn, impact the gut–brain axis. A previous randomised, double-blind, placebo-controlled trial was conducted to evaluate the effects of probiotic supplementation on cognition, physical activity, and anxiety in patients with mild to moderate AD [159]. Ninety eligible patients were randomly subjected to *Lactobacillus rhamnosus* HA-114 (10^15^ CFU), *Bifidobacterium longum* R0175 (10^15^ CFU), or a placebo. Each group received their assigned supplement twice daily for 12 weeks. After 12 weeks, significant improvement in Mini-Mental State Examination (MMSE) scores was observed, indicating enhanced cognitive function in the *Bifidobacterium longum* R0175 group compared to both the placebo group and the *Lactobacillus rhamnosus* HA-114. However, the two probiotic groups had no statistically significant difference in cognitive improvement.

In an in vivo study, transgenic 3xTg-AD mice were fed a probiotic diet containing *Lactobacillus plantarum* KY1032 and *Lactobacillus curvatus* HY7601 for 12 weeks [160]. Cognitive function was assessed through Barnes Maze trials, which showed improvements in memory performance among the probiotic-fed AD mice. Neural tissue analysis of the entorhinal cortex and hippocampus in 10-month-old 3xTg-AD mice revealed reduced astrocytic and microglial densities in the probiotic-supplemented group, with these reductions more prominent in female mice. Additionally, an increased number of neurons observed in the hippocampus of the probiotic-fed mice suggested a neuroprotective effect associated with probiotic supplementation. These findings indicate that probiotic supplementation may help delay or mitigate early neurodegenerative changes in the 3xTg-AD model. Another study examined the neuroprotective effects of the Lab4P probiotic consortium in 3xTg-AD mice, as well as in human SH-SY5Y cells [161]. In the mouse model, Lab4P supplementation helped prevent declines in novel object recognition, preserved hippocampal neuronal spine density, and reduced levels of the pro-inflammatory cytokines TNF-α and IL-1β in the hippocampus. In differentiated SH-SY5Y human neurons exposed to Aβ, Lab4P-derived metabolites demonstrated neuroprotective effects against Aβ and lowered the expression of IL-6.

Based on a meta-analysis study on the impact of probiotics on cognitive function in patients with MCI and AD, the analysis included six randomised controlled trials with a total of 462 MCI and AD patients [162]. The findings suggested that probiotic supplementation had a positive effect on homeostasis model assessment–insulin resistance (HOMA-IR) in AD patients. Probiotic administration also showed favourable effects on very low–density lipoprotein levels, the quantitative insulin sensitivity check index, and triglyceride levels in AD patients. However, after adjusting for statistical significance using the Hartung-Knapp method, only the effect on HOMA-IR remained significant. Changes in cognitive function, as measured by the MMSE and the Repeatable Battery for the Assessment of Neuropsychological Status, as well as biomarkers of oxidative stress, inflammation, and lipid profiles (such as high-sensitivity CRP, malondialdehyde, and total cholesterol), were minimal and non-significant.

These findings underscore the potential of microbiota-based therapies, particularly probiotics, as a supplementary approach for managing AD. While probiotics show promise in enhancing insulin sensitivity and potentially offering neuroprotective benefits, more study is needed to confirm these findings in larger and more diverse populations, determine optimal strains and dosages, and further elucidate the mechanisms by which probiotics impact the gut–brain axis in AD.

### 6.2. Faecal Microbiota Transplantation

FMT offers a promising approach to restoring gut microbiota diversity and mitigating inflammation associated with dysbiosis in AD. By reintroducing a balanced microbial community, FMT can increase beneficial bacterial populations, helping to restore gut microbiota equilibrium. A recent study using the TgCRND8 (Tg) AD mouse model investigated the impact of gut microbiota on AD pathology [163]. In this study, FMT was performed on young pseudo-germ-free (PGF) Tg mice (3 months old), with excessive choline supplementation introduced to examine how altered choline metabolism affects cognitive function. Results showed that transplanting dysbiotic gut microbiota from aged Tg mice into young PGF Tg mice accelerated AD pathology, evidenced by increased activation of the CDK5/STAT3 signalling pathway in the brain. In contrast, FMT from wild-type mice alleviated cognitive deficits in Tg mice, decreased neuroinflammatory markers (TNF-α, IL-6, and IL-17), reduced Aβ deposition and tau hyperphosphorylation, and lowered trimethylamine N-oxide levels in the brain tissue, while also suppressing CDK5/STAT3 pathway activation. These findings highlight the potential of FMT as a therapeutic strategy to modulate gut microbiota and counteract AD pathology. By rebalancing the gut microbiome, FMT may offer a pathway to reduce neuroinflammation and slow the progression of AD-related cognitive decline. While FMT has demonstrated potential in restoring gut microbial balance and attenuating neuroinflammation in AD models [163], its clinical translation faces significant challenges. The procedure’s invasiveness, risk of pathogen transmission, and variability in donor–recipient compatibility limit its widespread use. Additionally, long-term safety and efficacy in AD patients remain understudied.

### 6.3. Small-Molecule Therapies

Small-molecule therapies have gained considerable attention as a promising approach to modifying the underlying mechanisms of AD, particularly by targeting specific molecular pathways involved in amyloid plaque formation, tau phosphorylation, and neuroinflammation. These therapies consist of low molecular weight organic compounds (typically < 900 Daltons) that can easily penetrate cells and cross the BBB, allowing intervention at early stages of disease progression and potentially slow or halt cognitive decline associated with AD. One such small molecule, NaB, an HDAC inhibitor, has shown promise in preclinical studies for its ability to reduce Aβ accumulation and improve cognitive function. In 2020, Fernando et al. [122] demonstrated that NaB treatment alleviated memory deficits in the 5xFAD mouse model of AD following a 12-week feeding regimen [122]. The 5xFAD male mice were assigned to either a control diet or a NaB-supplemented diet at an early stage of disease (8–10 weeks of age). NaB treatment resulted in significant effects on both Aβ levels and cognitive performance, including a 40% reduction in brain Aβ levels and a 25% increase in fear responses during both cued and contextual testing, compared to the control group.

Cholinesterase inhibitors (ChEIs) represent the first generation of AD therapeutics, with tacrine being the initial FDA-approved agent in 1993 for mild-to-moderate AD [164,165]. While tacrine demonstrated modest cognitive benefits, its clinical utility was limited by hepatotoxicity, leading to the development of second-generation ChEIs including donepezil, galantamine, and rivastigmine [164,165]. Recent advances in ChEI design have focused on multi-target ligands, exemplified by compounds like ladostigil, which combines cholinesterase inhibition with MAO inhibitory activity and neuroprotective properties [166]. Novel structural modifications described by Adebambo et al. [167] have yielded ChEIs with enhanced selectivity for acetylcholinesterase and reduced peripheral side effects while maintaining cognitive benefits in preclinical models. Similarly, huperzine A, a natural ChEI derived from Chinese club moss, demonstrates additional antioxidant and anti-inflammatory effects that may address multiple AD pathways [168]. While these agents typically produce 2–3 point improvements in MMSE scores, their clinical impact remains limited to symptomatic relief without altering disease progression.

Memantine, an NMDA receptor antagonist approved for moderate-to-severe AD, represents another class of small-molecule therapeutics that modulates glutamatergic neurotransmission [169]. Other receptor-targeting approaches include α7 nicotinic acetylcholine receptor (nAChR) agonists such as EVP-6124 and GTS-21, which have shown promise in preclinical and early clinical studies by improving cognition and reducing neuroinflammation [169,170,171,172]. Significant efforts have focused on developing small molecules that target Aβ pathology through diverse mechanisms. BACE1 inhibitors like verubecestat (MK-8931) and AZD3293 significantly reduce Aβ production by inhibiting β-secretase cleavage of amyloid precursor protein (APP) [173]. However, clinical development has been challenged by mechanism-based side effects, including cognitive worsening and hepatotoxicity [173]. Alternative approaches targeting γ-secretase have evolved from non-selective inhibitors to notch-sparing modulators like CHF5074 and E2012 that preferentially reduce pathogenic Aβ42 while sparing essential Notch signalling [174].

Anti-aggregation strategies have yielded several promising small molecules, including tramiprosate (homotaurine) and its prodrug ALZ-801, which stabilise Aβ in non-toxic conformations [175]. Natural compounds like epigallocatechin gallate (EGCG) from green tea demonstrate potent anti-fibrillization activity through direct binding to Aβ oligomers [176]. Computational approaches have accelerated the discovery of novel aggregation inhibitors, with Han et al. [177] employing advanced *in silico* screening and molecular docking to identify small-molecule scaffolds with high binding affinity to Aβ oligomers.

Emerging research highlights the therapeutic potential of gut microbiota-derived metabolites and their synthetic analogues. Microbial Trp metabolites, particularly indole derivatives such as IPA, show potent antioxidant and anti-inflammatory properties that may protect against AD pathology [178]. Barresi et al. [179] have characterised several microbial metabolites with dual anti-amyloid and anti-inflammatory activities, highlighting their potential as lead compounds for AD drug development. Current research emphasises multi-target small molecules that simultaneously address multiple AD hallmarks. These hybrid compounds combine Aβ/tau targeting with antioxidant or anti-inflammatory activity, as demonstrated by recent work developing molecules with balanced pharmacodynamic profiles [179]. Small molecules remain essential in AD drug discovery due to their structural flexibility and ease of synthesis. When combined with insights from systems biology and gut microbiome studies, they provide powerful avenues for developing novel therapeutic strategies. Despite ongoing challenges in optimising pharmacokinetic properties and achieving consistent clinical efficacy, small-molecule therapies continue to hold a central position in the AD treatment landscape. Importantly, they offer promising opportunities for integration into combination therapies and for advancing personalised medicine approaches tailored to individual patient needs.

### 6.4. Protein-Peptide Drug Therapies

Monoclonal antibodies (MABs), protein-based therapeutics, have emerged as a promising approach for AD treatment, offering high specificity in targeting pathological protein aggregates. MAB represent the most advanced class of these biologics, with lecanemab and aducanumab receiving FDA approval for their ability to target Aβ species through distinct mechanisms. Lecanemab preferentially binds soluble Aβ protofibrils while aducanumab recognises aggregated forms, both promoting clearance through Fc receptor-mediated phagocytosis by microglia [180]. Clinical trials demonstrated these antibodies can slow cognitive decline by 27–32% as measured by CDR-SB scores, accompanied by a significant reduction in amyloid PET signal [180]. However, their clinical application requires careful monitoring due to the risk of amyloid-related imaging abnormalities (ARIA), which occur in 12–17% of treated patients. Donanemab, which targets pyroglutamate-modified Aβ, has shown particularly robust plaque clearance in phase II trials, especially in patients with high tau burden [180].

Several studies have expanded the scope of protein therapeutics to include microbiome-derived proteins that modulate neuroinflammation through the gut–brain axis. Moving beyond traditional probiotics, specific bacterial proteins are being isolated for their direct therapeutic effects. Ayan et al. [181] employed a metaproteomic approach to identify beneficial proteins from commensal bacteria such as *Bacteroides* and *Faecalibacterium* species in a transgenic AD mouse model. Their study identified several key microbial proteins, typically ranging from 10–50 kDa, that are associated with improved AD phenotypes. For instance, proteins such as glutathione synthase (from *Bacteroides*) and CoA-transferase (a key enzyme for butyrate production from *Faecalibacterium prausnitzii*) were highlighted [181]. These microbial proteins function by activating the peroxisome proliferator-activated receptor-gamma (PPAR-γ) signalling pathway, a key regulator of inflammation and metabolism [181]. This activation leads to a reduction in pro-inflammatory cytokines like IL-6 and TNF-α, while simultaneously enhancing the expression of neuroprotective factors. Furthermore, this modulation of PPAR-γ may contribute to the maintenance of BBB integrity by strengthening tight junction proteins, thereby reducing the influx of peripheral inflammatory molecules into the brain [181]. This offers a novel, targeted approach to complement existing AD therapies by directly leveraging the anti-inflammatory and barrier-protective properties of commensal gut bacteria.

Concurrently, therapeutic peptides represent a promising class of protein-based treatments with several advantages over larger antibodies, including superior tissue penetration, reduced immunogenicity, and the ability to be chemically synthesised with high purity. A primary strategy involves developing short peptide inhibitors designed to disrupt the pathological aggregation of Aβ and tau β-sheet breaker peptides. For example, β-sheet breaker peptides like iAβ5 attach to Aβ and stop it from forming the sticky β-sheet structures that lead to toxic oligomers and fibrils [182]. The findings demonstrated that engineered β-hairpin peptides that mimic the critical recognition epitopes of Aβ can effectively block oligomerisation at sub-stoichiometric ratios, thereby neutralising these toxic species before they can impair synaptic function [182]. Recent advances in peptide engineering have focused on overcoming the inherent limitations of natural peptides, such as poor metabolic stability and low bioavailability. Strategies include the development of stable analogues resistant to proteolysis (e.g., through D-amino acid substitution or cyclisation) and the creation of cell-penetrating peptides (CPPs) to improve delivery across the BBB [183,184]. The frontier of this field involves designing multifunctional peptides that combine Aβ/tau recognition sequences with neuroprotective motifs. For example, peptides may be conjugated to antioxidant moieties (e.g., catalase mimics) or anti-inflammatory sequences, creating single compounds that simultaneously target multiple disease mechanisms, including aggregation, oxidative stress, and neuroinflammation [185,186].

### 6.5. Dietary Interventions

The associations between diet and neuroprotection, along with cognitive health, have garnered considerable attention, with specific dietary patterns and nutrient supplementation explored as potential strategies for reducing the risk of AD. The MeDi is defined as a dietary pattern that promotes the intake of bread, legumes, vegetables, fruits, and unsaturated fats (olive oil, nuts, and seeds). The diet comprises moderate quantities of fish and poultry, limits dairy and meat consumption, and typically includes wine in moderation with meals. A previous study indicated that adherence to the MeDi may lower the risk of AD.

A longitudinal cohort study examined the relationship between adherence to the MeDi and cognitive outcomes [187]. This study included 1393 cognitively normal individuals, who were observed for an average of 4.5 years and classified into three tertiles based on their adherence to the MeDi. Compared to the lowest adherence group, individuals in the middle tertile showed a non-significant 17% reduction in the risk of developing MCI. Participants in the highest adherence tertile demonstrated a statistically significant 28% reduction in the risk of MCI. A parallel analysis of 482 participants with MCI over an average follow-up of 4.3 years indicated that individuals in the middle tertile of the MeDi adherence exhibited a statistically significant 45% reduction in the risk of developing AD compared to those in the lowest tertile, while those in the highest tertile showed a 48% lower risk. The results demonstrate that increased adherence to the MeDi is associated with a reduced risk of MCI and a notably lower likelihood of progression from MCI to AD, underscoring the cognitive benefits of this dietary approach.

Furthermore, the systematic review consolidates findings from 32 studies involving 25 distinct cohorts, including five randomised controlled trials (RCTs) and 27 observational studies, that examine the association between adherence to the MeDi and cognitive outcomes [188]. Research demonstrates that greater adherence to the MeDi is associated with improved cognitive function, reduced risk of cognitive impairment, and a decreased likelihood of developing dementia, including AD. Nevertheless, there is an inconsistency noted. The analysis also finds that three studies reported no association between adherence to the MeDi and AD, three found no link with cognitive impairment, and five indicated no relationship with cognitive function. The studies reviewed exhibited variability in design, quality, and outcomes, resulting in heterogeneity in the findings. The cumulative data indicate a correlation between adherence to the RCTs and improved cognitive performance. However, causality is not established due to the prevalence of observational studies. Further rigorous controlled trials are necessary to validate these findings and elucidate the role of MeDi in the prevention or reduction of the effects of cognitive decline.

The modified Atkins diet (MAD), a variation of the KD, has also been associated with AD. This diet is defined by a high-fat and low-carbohydrate composition, where carbohydrates account for roughly 10% of total energy intake. This dietary method shifts the body’s energy substrate from glucose to ketone bodies, including acetoacetate and β-hydroxybutyrate. This metabolic adaptation is akin to fasting and provides an alternative energy source for the brain, potentially benefiting conditions like AD. A Phase I/II randomised clinical trial evaluated the feasibility and effects of a MAD on ketone production and memory in individuals with MCI or early AD [189]. Of the 27 participants recruited over 2.5 years, 14 completed the study, comprising nine in the MAD group and five in the control group, both adhering to the National Institute on Ageing (NIA) diet. Adherence levels were moderate across both groups. In MAD participants exhibiting detectable urinary ketones, episodic memory showed a significant improvement at six weeks.

A distinct study examined a medium-chain triglyceride (MCT)-supplemented KD in individuals with clinical dementia ratings between 0.5 and 2 [190]. Of the 15 participants enrolled, 10 completed the trial successfully. The intervention resulted in improved cognitive performance, demonstrated by a 4.1-point increase in scores on the Alzheimer’s Disease Assessment Scale-Cognitive Subscale. However, these enhancements reverted to baseline after a washout period. Adverse events, primarily linked to MCTs and career burden, resulted in withdrawals, particularly among participants with advanced dementia. The results indicate that KD may improve cognitive outcomes in early AD. However, they highlight the adherence and safety challenges that necessitate further research. In AD, alterations in lipid composition and membrane degradation partially influence synaptic and neuronal losses. Phospholipids, such as phosphatidylcholine and phosphatidylserine, exhibit potential for improving cognitive function. The OmegAD study, a clinical trial investigating omega-3 supplementation in individuals with mild-to-moderate AD, showed cognitive preservation, reduced inflammation, and increased plasma and cerebrospinal fluid levels of DHA and EPA following a six-month duration [191]. However, the evidence regarding the effectiveness of omega-3 supplementation in preventing cognitive decline in the general elderly population is limited.

The PREDIMED and FINGER trials, two landmark studies in dementia prevention, provide compelling evidence supporting the long-term benefits of lifestyle-based interventions on cognitive health in at-risk elderly populations [192,193]. The PREDIMED-Navarra and PREDIMED-Barcelona trials, part of the broader PREDIMED study, assessed the effects of MeDi supplemented with either extra virgin olive oil (EVOO) or nuts in individuals at high vascular risk. Over 4 to 6.5 years of follow-up, both trials demonstrated significant improvements in global cognition, memory, and executive function, with lower rates of MCI observed, particularly in the MeDi + EVOO group. These cognitive benefits were independent of key confounders such as age, sex, APOE ε4 status, and vascular risk factors. Complementing these findings, the FINGER trial, a RCT conducted in the general Finnish population, examined the effects of a long-term multidomain lifestyle intervention including diet, exercise, cognitive training, and vascular risk monitoring. Despite modest effect sizes (Cohen’s d = 0.13), the study demonstrated meaningful cognitive benefits over two years, emphasising the public health significance of even small cognitive gains in delaying the onset of dementia. With extended follow-ups underway in both trials to assess dementia and Alzheimer’s incidence, these studies collectively underscore the potential of holistic, non-pharmacological strategies in mitigating cognitive decline and offer scalable, evidence-based models for global dementia prevention initiatives.

**Table 1 ijms-26-08905-t001:** Therapeutic strategies targeting the gut–brain axis in Alzheimer’s disease.

Therapeutic Approach	Key Effects/Mechanisms	References
Probiotics	Improved cognitive function; Reduced neuroinflammation; Modulated microbial composition	[159,160,161]
Faecal Microbiota Transplantation	Restored microbial diversity; Reduced Aβ/tau pathology; Decreased neuroinflammation	[163]
Small molecules		
Cholinesterase inhibitors	Enhanced cholinergic transmission; Symptomatic cognitive improvement	[164,165,166,167,168]
BACE1/γ-secretase modulators	Reduced Aβ production; Selective APP processing	[173,174]
Anti-aggregation compounds	Inhibited Aβ oligomerization/fibrillization	[175,176,177]
Gut-derived metabolites	HDAC inhibition (butyrate); Antioxidant/anti-inflammatory effects (indoles)	[122,178,179]
Protein-peptide therapies		
Monoclonal antibodies	Targeted Aβ clearance; Reduced plaque burden	[180]
Microbial proteins	Modulated neuroinflammation; Protected BBB integrity	[181]
Anti-aggregation peptides	Inhibited Aβ/tau aggregation; β-hairpin scaffolds	[182]
Dietary Interventions		
Mediterranean diet	Improved cognitive function; Reduced AD risk	[187,188]
Ketogenic diet	Enhanced ketone metabolism; Reduced amyloid burden	[189,190]
Omega-3 supplementation	Anti-inflammatory effects; Preserved neuronal function	[191]

## 7. Current and Emerging Therapeutic Strategies for Alzheimer’s Disease: From Small Molecules to the Microbiome

Currently available pharmacologic interventions for AD, particularly small-molecule therapies such as ChEI (donepezil, rivastigmine) and the NMDA receptor antagonist memantine, offer only limited symptomatic improvement, without having a meaningful impact on the underlying progression of the disease [194]. Newer MABs, such as aducanumab, lecanemab, and donanemab, have demonstrated statistically significant effects in slowing cognitive decline [195]. For instance, lecanemab reduced clinical decline by 27% compared to placebo in the Clarity AD trial over 18 months [196]. However, the absolute difference in the Clinical Dementia Rating–Sum of Boxes (CDR-SB) was only 0.45 points, raising questions about clinical meaningfulness. Moreover, these therapies are associated with serious safety concerns, including ARIA such as brain swelling or microhemorrhages, occurring in 12–17% of treated individuals.

Beyond approved treatments, a wide range of experimental small-molecule therapies targeting AD pathophysiology have been explored, particularly those aimed at Aβ and tau pathways. BACE1 inhibitors, such as Verubecestat, LY2811376, and AZD3293, successfully lowered Aβ levels in preclinical and early clinical studies [197]. However, these agents failed in later trials due to toxicity or lack of cognitive efficacy, with some even worsening cognitive outcomes [196]. γ-secretase inhibitors like Semagacestat also failed because of off-target toxicities related to Notch signalling [198]. More selective γ-secretase modulators, such as CHF5074, showed some promise, but further evidence is needed to confirm these benefits, particularly in humans [199]. Similarly, aggregation inhibitors like tramiprosate, ALZ-801, and EGCG have shown biochemical or preclinical effects, yet few have translated into clinical efficacy [198]. Tau-targeted agents such as LMTX have demonstrated potential in monotherapy trials, but results remain inconclusive.

In contrast, gut microbiota-based and dietary interventions remain in earlier stages of development but offer intriguing promise. Several small-scale studies suggest that probiotics and FMT can beneficially affect cognition, inflammation, and gut–brain axis signalling. For example, a 12-week probiotic trial in AD patients [199] reported a 1.2-point improvement in MMSE scores compared to controls, alongside reductions in inflammatory markers. Several studies are also focusing on bioactive molecules derived from gut microbiota, such as proteins, peptides, and small molecules, which may modulate neurodegenerative processes or dysbiosis. These compounds represent promising targets for screening and therapeutic development [200]. Besides that, other molecular pathways should also be considered, particularly the interactions between gut microbiota and the host. Specifically, the crosstalk between gut microbiota-derived metabolites and host secretome plays a crucial role in maintaining homeostasis. Disruptions in gut microbiota composition may impair this interaction, thereby altering the secretome profile (including non-coding RNAs, cytokines, enzymes, and hormones), which in turn can influence disease progression. Such insights could pave the way for personalised interventions, addressing the clinical heterogeneity of AD and offering strategies for early prevention. FMT studies in animals have demonstrated reductions in Aβ and tau pathology and improvements in cognitive function, while pilot human studies report anecdotal cognitive gains and altered gut microbiota profiles [160]. However, these trials are often limited by small sample sizes, lack of biomarker validation, and absence of long-term follow-up or placebo controls. When compared to dietary interventions like the MIND diet, which has been associated with up to a 53% reduced risk of developing AD and slower cognitive decline (hazard ratio ~ 0.47), gut microbiota-based approaches show similar potential but require much stronger clinical evidence [201].

A large-scale, rigorously designed dietary intervention trials like PREDIMED and FINGER provide robust evidence supporting the cognitive benefits of lifestyle-based strategies [192,193]. The PREDIMED trials demonstrated that adherence to a MeDi supplemented with extra virgin olive oil or nuts improved global cognition and memory in older adults at high vascular risk, with benefits sustained over 4 to 6.5 years. Similarly, the FINGER trial showed that a multidomain lifestyle intervention, including diet, exercise, cognitive training, and vascular risk management, produced significant cognitive improvements in at-risk elderly individuals over two years, with follow-up studies underway to assess long-term effects on dementia incidence. Compared to gut microbiota-based interventions, these dietary and lifestyle approaches offer more established and replicable benefits, supported by large sample sizes, validated cognitive assessments, and extended follow-ups. Nevertheless, gut microbiota-based strategies may offer complementary or synergistic effects and, if validated in well-powered, biomarker-driven trials, could further expand the non-pharmacological toolkit for AD prevention and cognitive health.

## 8. Future Direction

AD is a progressive neurodegenerative condition characterised by complex and heterogeneous pathophysiology involving overlapping mechanisms such as Aβ accumulation, tauopathy, neuroinflammation, mitochondrial dysfunction, and alterations in the gut–brain axis. This heterogeneity manifests not only in biological underpinnings but also in clinical symptoms, progression rates, and treatment responses, with some individuals experiencing prominent memory loss and others presenting behavioural or language impairments. Compounding this complexity, mixed pathologies such as cerebrovascular disease, Lewy bodies, or TDP-43 proteinopathy are commonly observed in the elderly and further hinder accurate diagnosis and therapeutic efficacy.

These diverse and often overlapping pathological features underscore a critical distinction in therapeutic strategy. The interventions that aim to reverse AD symptoms or biomarkers must confront the challenge of restoring damaged neurons and disrupted brain circuitry, which becomes increasingly infeasible as the disease progresses. In contrast, therapeutic approaches designed to slow or arrest further development of AD, especially when applied during early or even preclinical stages, offer more viable opportunities to preserve cognitive function and delay symptom onset. This has profound implications for the timing and targets of treatment efforts. What we need to do better is to enhance early detection capabilities, allowing for earlier intervention before irreversible neural damage occurs, and to design interventions that consider the full spectrum of AD heterogeneity, including co-existing pathologies and individual variability in disease progression.

Precision medicine strategies are essential for advancing treatment effectiveness in AD. These strategies stratify patients based on biomarker profiles, comorbidities, and disease stages. Improved diagnostics, especially those that can be applied early and non-invasively, along with investment in long-term mechanistic research, are essential steps forward. Future therapeutic strategies should prioritise combination approaches targeting upstream disease mechanisms such as inflammation, metabolism, and gut microbiota dysfunction before neurodegeneration becomes irreversible. A growing body of evidence highlights the critical role of gut microbiota in AD, particularly through its influence on inflammation and neurodegeneration. Understanding how microbiota composition varies across regions of the gastrointestinal tract and how it evolves over the stages of AD may reveal early biomarkers and windows of intervention. Specific microbial strains and metabolites, such as SCFAs, bile acids, and microbial amyloids, may either promote neuroprotection or exacerbate neuroinflammation. Additional key factors, such as protein aggregation states, especially those involving Aβ and tau, and redox imbalance in the brain, must be addressed more explicitly. These pathological features are closely linked to both mitochondrial dysfunction and oxidative stress, which can be directly influenced by gut-derived metabolites. Small molecules targeting these pathways, including those that modulate mitochondrial function, reduce oxidative stress, or influence the immune system via gut microbial signalling, are promising adjunct therapies.

Integrating these dimensions, protein aggregation, redox status, immune modulation, and microbial metabolism, through multi-omic approaches (e.g., metagenomics, metabolomics, transcriptomics, and proteomics) is essential for identifying precise biomarkers and therapeutic targets. These tools allow for a systems-level understanding of the gut–brain axis, capturing its multifaceted role in AD pathology. Therapeutic interventions aimed at gut microbiota, including probiotics, FMT, dietary modifications, and small-molecule therapies, present potential strategies for minimising AD pathology by re-establishing microbial equilibrium, decreasing inflammation, and enhancing brain health. Further research is necessary to clarify the mechanisms of these interactions, pinpoint specific microbial biomarkers and therapeutic targets, and create tailored strategies for prevention and treatment. The integration of advanced multi-omic approaches, alongside the consideration of individual and population-level differences, significantly enhances the potential to mitigate cognitive decline and improve outcomes for patients with AD, enabling therapeutic approaches diversified with respect to clinical features and progressive conditions.

## Figures and Tables

**Figure 1 ijms-26-08905-f001:**
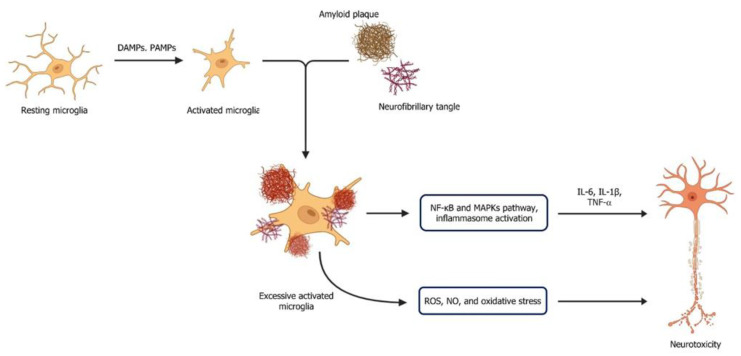
The mechanism of microglial activation and neurotoxicity. It involves the activation of resting microglia by DAMPs and PAMPs, resulting in their interaction with amyloid plaques and NFTs. Thereafter, it activates the NF-κB and MAPK pathways, as well as inflammasome activation, resulting in the production of pro-inflammatory cytokines (TNF-α, IL-1β, and IL-6). The overactivation of microglia leads to the generation of ROS and NO, which induce oxidative stress and neurotoxicity.

**Figure 2 ijms-26-08905-f002:**
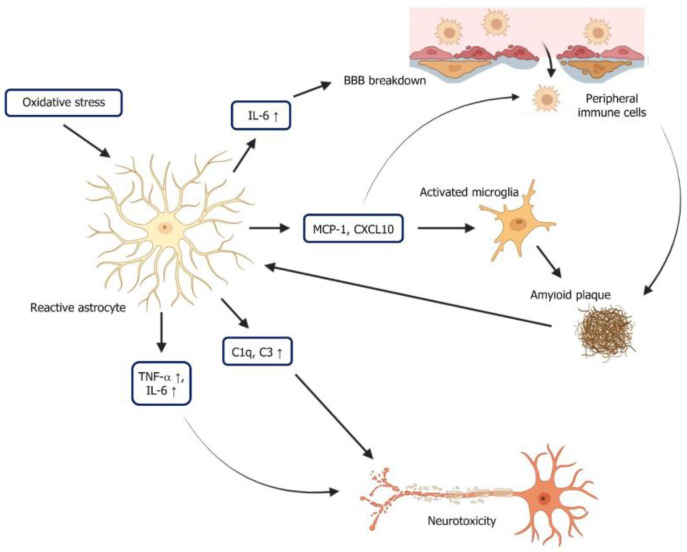
Activation of astrocytes resulting in neurotoxicity: Oxidative stress and amyloid plaques stimulate reactive astrocytes to release TNF-α, IL-6, C1q, and C3. Elevation of these mediators contributes directly to neuronal damage. Excessive IL-6 production leads to BBB collapse, permitting systemic inflammation to infiltrate the brain. Astrocytes synthesise MCP-1 and CXCL10, which activate microglia and facilitate peripheral immune cells in containing amyloid plaques. This pathway promotes and amplifies inflammation in the brain, consequently leading to neurotoxicity and contributing to neuronal damage.

**Figure 3 ijms-26-08905-f003:**
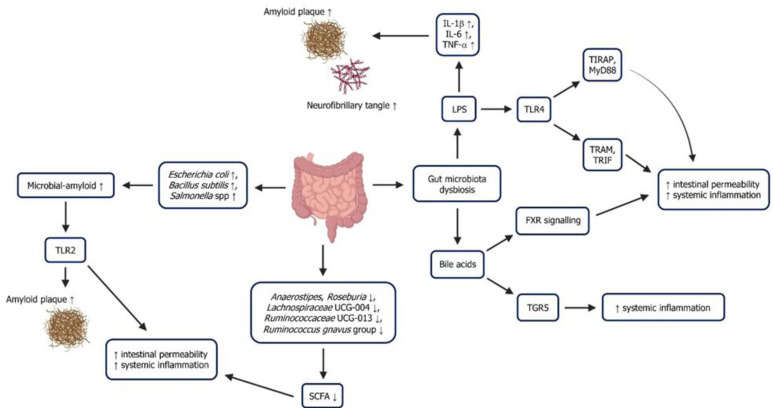
The correlation between gut microbiota dysbiosis and inflammation: Dysbiosis of the gut microbiota leads to elevated concentrations of LPS and microbial amyloids. An increased abundance of *Escherichia coli*, *Bacillus subtilis*, and *Salmonella* spp. enhances microbial amyloid production, which activates TLR2 and promotes amyloid plaque formation in the brain. At the same time, LPS activates TLR4 pathways (via TIRAP/MyD88 or TRAM/TRIF), increasing intestinal permeability and driving systemic inflammation. In parallel, it elevates pro-inflammatory cytokines (IL-1β, IL-6, and TNF-α), further accelerating the formation of amyloid plaques and NFTs. Dysbiosis also alters bile acid metabolism and exacerbates systemic inflammation through the FXR/TGR5 signalling pathway. Conversely, a reduced abundance of beneficial bacteria such as *Anaerostipes*, *Roseburia*, *Lachnospiraceae* UCG-004, *Ruminococcaceae* UCG-013, and the *Ruminococcus gnavus* group lowers SCFA production, further compromising intestinal barrier integrity and amplifying systemic inflammation. Collectively, these mechanisms lead to elevated soluble Aβ, phosphorylated tau, amyloid plaques, NFTs, and persistent systemic neuroinflammation.

## Data Availability

Not applicable.

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
