# Peer review of "The Interplay of Inflammation and Gut-Microbiota Dysbiosis in Alzheimer’s Disease: Mechanisms and Therapeutic Potential"

_ijms, 2025, doi:10.3390/ijms26188905_

Round 1
Reviewer 1 Report
Comments and Suggestions for Authors
You have provided an overview for the link between gut microbiota (GMB), inflammation and Alzheimer's Disease. A good review is not just a compilation of available data but ideally should provide critical commentary and some improvement on already available reviews. In other words, you should ask yourself - what is the incremental value of our review?
I would suggest by considering some recent reviews that your review will be competing with for attention. Can you improve on those? Should you integrate any of their comments?
Gut microbiota in Alzheimer's disease: Understanding molecular pathways and potential therapeutic perspectives
Review Ageing Res Rev . 2025 Feb:104:102659.
doi: 10.1016/j.arr.2025.102659. Epub 2025 Jan 10.
PMID: 39800223 DOI: 10.1016/j.arr.2025.102659
IUPHAR review: From gut to brain: The role of gut dysbiosis, bacterial amyloids, and metabolic disease in Alzheimer's disease
Review Pharmacol Res . 2025 May:215:107693.
doi: 10.1016/j.phrs.2025.107693. Epub 2025 Mar 12.
PMID: 40086611 DOI: 10.1016/j.phrs.2025.107693
Gut microbiota changes in patients with Alzheimer's disease spectrum based on 16S rRNA sequencing: a systematic review and meta-analysis
Front Aging Neurosci. 2024 Aug 8:16:1422350.
doi: 10.3389/fnagi.2024.1422350. eCollection 2024.
PMID: 39175809 PMCID: PMC11338931
Provide a critical assessment of our current success using GMB-mediated approaches. Is it strong? weak? absent?
Of the approaches that you mention, the most consistent and largest effect size is dietary, particularly for the MED diet. While many are by association (e.g. population studies) are there some that are prospective? You mention SCFAs. Are there other small molecules that could be implicated?
AD is a condition with great mechanistic heterogeneity and generally multi-year progression. What are the implications for identifying therapeutic approaches that reverse AD symptoms/biomarkers v those that either slow or arrest further development? What do we need to do better?
Provide more context - What is the success (e.g. effect sizes or hazard ratio) of currently available pharmacologic interventions? How do those compare to the dietary or GMB-based interventions you review?
Poorly substantiated statements: "endogenous GABA generated in the gut can bypass the intestinal epithelia, enter the lamina propria, and circulate in the circulation to the brain."
If you have a reference that strongly supports that claim, provide it. If the statement is controversial or speculative, indicate that.
The contribution of GMB-derived GABA to circulating GABA in the systemic circulation
Stylistic comments
Italicize terms such as Eubacterium rectale and and Bacteroides fragilis
Correct incomplete sentences such as : Based on a meta-analysis study on the impact of probiotics on cognitive function in 592 patients with MCI and AD [113].
Correct spelling errors: nuerotransmitter
Comments on the Quality of English Language
Correct incomplete sentences such as : Based on a meta-analysis study on the impact of probiotics on cognitive function in 592 patients with MCI and AD [113].
Correct spelling errors: nuerotransmitter
Reviewer 2 Report
Comments and Suggestions for Authors
The narrative review written by Samat et al, entitled: "The Interplay of Inflammation and Gut-Microbiota Dysbiosis in Alzheimer's Disease: Mechanisms and Therapeutic Potential", deals with a largely investigated hypothesis of Alzheimer's Disease (AD) progression: that focused on the role of gut microbiota and gut-brain axis. Despite broad consensus in the scientific community on the assessment of patients' microbiomes and their variability, this field of research is still in its early stages, which underlines, given the severe disability and invalidity of this neurodegenerative disease, the need to propose new assessment and intervention models, considering this aspect too. On the other hand, there are already many papers on this topic in the current literature, and further reviews in the field should be more and more exhaustive and original in respect to the past published works. Overall, the review of Samat et al. is clearly and correctly written, but it requires new content. Thus, the issue must be addressed by considering all points of view, highlighting any limit and bias, in order to delve deeper into the topics presented.
Abstract
Some few suggestions are proposed.
Lines 35-36: please check the sentence and modify accordingly to this example, preferring a more measured tone in the presentation: "This imbalance is thought to be associated with alterations in the concentrations of short-chain fatty acids (SCFAs) and bile acids, which can modulate neuroinflammation and contribute to AD pathology".
Lines 38-40: Considering the previous sentence, please also check this one accordingly: " This review provides an overview of the hypothesis that systemic and CNS inflammation together with gut microbiota dysbiosis may interact to influence the development and progression of AD."
Introduction
A brief but comprehensive section on the epidemiology of AD would increase the scientific impact of this review. Indeed, epidemiology provides fundamental information, such as the global spread of this highly debilitating disease, its incidence and prevalence, its impact on the elderly, and its evolution in the coming years. In addition, one aspect that may be relevant to the topic presented is that related to the different lifestyles of older people, the presence of biological vulnerability and/or previous diseases and health conditions that could promote AD. Nutritional aspects and stress-related contexts should be taken into account. The Introduction section should provide a starting point for subsequent discussion questions, offering a broad overview of all the biological and neurobiological aspects potentially involved. For example, authors should also consider articles such as the following: "Dhana K, Evans DA, Rajan KB, Bennett DA, Morris MC. Healthy lifestyle and risk of Alzheimer's dementia: results from 2 longitudinal studies. Neurology. 2020;95(4):e374-e383; Xu Lou I, Ali K, Chen Q. Effect of nutrition in Alzheimer's disease: a systematic review. Front Neurosci. 2023;17:1147177; Madore C, Yin Z, Leibowitz J, Butovsky O. Microglia, lifestyle stress, and neurodegeneration. Immunity. 2020;52(2):222-240; Bravo-Jimenez, M.A., Sharma, S. Karimi-Abdolrezaee, S. The integrated stress response in neurodegenerative diseases. Mol Neurodegeneration 2025; 20: 20. " Indeed, these aspects may shape species diversification of gut microbiota.
Gene x environment aspects must also be considered, as these introduce the role of the combined action of microbiota with host genetics and the interaction of its diversity on human health, representing the molecular basis of cooperation between microbial and human metabolism. The colonization of the gut microbiota is also one of the main events in early life that influence the immune response and, therefore, the reactivity of the immune system. Inflammation is mentioned in this paper, but the molecular mechanisms of inflammation in Alzheimer's disease and the link between the immune system and the inflammatory response need to be presented and then discussed in greater depth in order to introduce the potential role of the microbiota (Wu KM, Zhang YR, Huang YY, Dong Q, Tan L, Yu JT. The role of the immune system in Alzheimer's disease. Ageing Res Rev. 2021 Sep;70:101409).
Main sections in the manuscript
The body of this review should therefore be modified and expanded, along the above suggested lines.
Alzheimer's disease should be described in more detail, with its various signs and symptoms and stages of disability, highlighting its heterogeneity.
As well, any other aspect presented here should also be explored in greater depth. The permeability of the blood-brain barrier has been considered but other molecular patterns, as the CypA-MMP9 pathway or the NF-kB one must be reported more comprehensively (Alkhalifa AE, Al-Ghraiybah NF, Odum J, Shunnarah JG, Austin N, Kaddoumi A. Blood-Brain Barrier Breakdown in Alzheimer's Disease: Mechanisms and Targeted Strategies. Int J Mol Sci. 2023;24(22):16288.
The roles of macromolecules and that of small molecules in this devastating illness should be better defined. Protein aggregation state and redox conditions require to be addressed more specifically. As regards small molecules, the role of tryptophan metabolism should be indicated for its link with immune response and gut microbiota metabolism. This would allow for a better understanding of the usefulness of the -Omic approach proposed by the authors, involving AD therapeutic improvement and prevention. Essentially, the role of the gut-brain axis in AD must be assessed thoroughly, taking into account all its facets.
Others: The authors should check the reference list: many citations do not indicate the full list of authors' names but just the first name followed by et al. It is always preferable to adopt consistent criteria and to indicate all authors.
Comments on the Quality of English LanguageThe quality of English language is correct. The text requires to be carefully revised, and, therefore, obviously additional English check.
Round 2
Reviewer 1 Report
Comments and Suggestions for Authors
The manuscript ha been considerably improved.
Author Response
Comment 1: The manuscript has been considerably improved.
Response 1: We sincerely thank the reviewer for their thoughtful evaluation and for recognizing the improvements made to the manuscript. We are pleased that the revised version meets their expectations and appreciate the time and expertise they have contributed to enhancing the quality of our work.
Reviewer 2 Report
Comments and Suggestions for Authors
The new version of the manuscript ijms-3723397, entitled: "The Interplay of Inflammation and Gut-Microbiota Dysbiosis in Alzheimer's Disease: Mechanisms and Therapeutic Potential" shows improvements and the overall presentation of the work is clearer, as suggested. However, there appear now other issues or problems with the additional parts introduced to improve the manuscript based on the first review, here summarized.
Page 2, line 95: At this point, the authors should describe the microbiota-host interactions; precisely, by indicating that the gut microbiota colonizes the human intestine from the earliest stages of life, establishing a close bidirectional relationship with the host: the variety of microbial species changes during the various stages of host's growth and development, adapting to host lifestyle and diet, providing essential elements for human metabolism and receiving nutrients and substrates in return. The host, on the other hand, can modify its metabolism thanks to advantageous changes in the microbiota variety. The relationship is therefore strong, given the reciprocity of coexistence between humans and microorganisms. The microbiota also provides a sort of immunological “model” to the host, strengthening its immune system and initiating innate immunity mechanisms; for this reason, authors should consider, among others, the article by Xiao et al, Microbe-host interactions: structure and functions of Gram-negative bacterial membrane vesicles. Front Microbiol. 2023;14:1225513, for its introductory and explanatory sections.
Page 3, lines 105-108: at line 107, the authors should change similarly to: "..adjustable risk factors in the prevention and progression of many relevant diseases, including neurodegenerative ones, as dementia and AD"
Page 4, lines 179-180: The authors should avoid the term race and use a more scientific language; they should mention only : "..age, gender, population and ethnicity". The term race is not accepted anymore by the scientific community. Furthermore, authors should preferably use the term gender, which is more appropriate in demographic reports. It is also a term that better respects the different situations relating to gender identity.
Page 5, Line 192-193: "Ethno-racial..": As afore mentioned, the authors should avoid the suffix “racial,” given the historically unethical and discriminatory use of this term, even if authors are surely acting in good faith. They should prefer simply the word “ethnic,” a term accepted by the scientific community. For instance, they should change the sentence as that:" Ethnical vulnerabilities defining the risk of developing AD and related dementias have been also reported".
Page 5, lines 193-197: These sentences should be also carefully revised: epidemiological reports should be up-to date and recent. Moreover, the authors should avoid the expression " Older Black/African American adults", rather they should use "elderly African-Americans"; the same applies for "older white adults", rather preferring: "elderly people of European origin "
Also: the authors should verify this sentence by new and recent references for lines 195-197, preferably using the term "people from Latin America".
Page 6, line 264: The authors should briefly explain what is TSPO, its function and why this specific molecular target has been chosen in these studies.
There is also no introductory paragraph on pages 12-13 or introduction on microbiota and gut microbiota in general, e.g. the main colonizing species, and the microbiology of these species. To write this part, the authors should consider and cite a very recent article from 2024/2025. A figure, diagram or Table would also be appreciated at this point. This is important for the article exhaustiveness. The review should possible contain new information because the topic is widely covered and, at the same time, it must be highly up to date.
Page 19, line 852: The authors should mention the limits of this therapeutic intervention, and its invasiveness. They should try to search articles containing perspectives for new future developments and methodologies in this field. The authors should add information about other less invasive modulation strategies of gut microbiota (probiotics, prebiotics, others) in AD, possibly reported in a new paragraph.
Also the organization of the manuscript could be ameliorated, following a more linear trend. I suggest, for instance:
Paragraph 2: Alzheimer's Disease
2.1. Main clinical, anatomical and histological features
2.2. Demographic features
2.3 Heterogeneity of Alzheimer's Disease.. etc..
The title should be revised as, for instance: 5. Gut-microbiota dysbiosis, inflammation and Alzheimer's Disease
Comments on the Quality of English Language
The scientific language should be still revised carefully, sometime there are imprecise and somehow inappropriate statements..
Round 3
Reviewer 2 Report
Comments and Suggestions for Authors
The third version of the article written by Samar et al, IJMS ID -3723397 , has been now provided. The current manuscript is much improved than previous ones. Indeed, the authors have carefully revised their article based on all the given suggestions, the text has been expanded as requested, and this latest version follows a much more precise outline. However, as for the second version, other critical points are now emerging or have been introduced in the newly added sections. Some sections are still poorly treated. Therefore, more efforts are needed to achieve a well-focused narrative review on a highly challenging medical issue such as the pathophysiology and treatment of a devastating neurodegenerative disease such as Alzheimer's disease.
General comments: the authors should check the names of microorganisms through the manuscript, and Italic characters should be used. For phyla names too, as presented in many papers on this topic. More: English language still require revision.
Specific comments:
Introduction
Lines 67-69, page 2: this sentence is still unclear; this alternative one is, for instance, more precise: “The incidence and progression of AD are increasingly recognised as being impacted not just by neuropathological changes but also by individual lifestyle choices, overlapping biological predispositions, and pre-existing health disorders. “
Lines 114-126, page 3: Authors should review the English language of this part of the manuscript for a more rigorous presentation, for example:” This review aims to resume the main current understanding of gut microbiota dysbiosis and related metabolism in respect to pathways of central and systemic inflammatory responses..These factors contribute..Altered Inflammation processes have emerged as central components.. Therefore, this narrative review also explores..By elucidating key…this paper intends to focus on current understanding..for the prevention, treatment and monitoring of AD. “
Paragraph 3
Lines 269-275, page 6, on TSPO: the role of TSPO needs to be better explained, indicating briefly the involvement of this protein in mitochondrial function, energy production, ROS generation, cholesterol transport and steroidogenesis: this point has not be clearly stated in the current form.
Paragraph 4 , page 11: This paragraph is too short, it should be extended by authors by considering and citing the paper : Hou et al , Microbiota in health and disease, Signal transduction and targeted therapy 2022; 7:135. This is a general and comprehensive review, very useful in the state-of-the-art of the topic, including gut microbiota. This fourth section should be at least double its current size, without containing any discussion of the gut microbiota in Alzheimer's disease. In fact, this is just a general section, which provides the rationale for exploring the gut microbiota in Alzheimer's disease. It's a sort of introductory subsection to the main topic of this article.
Page 20. 7.3. Small-molecule therapies. this sub-paragraph lacks some important issues, related to the whole aim of the paper: at Line 933 the authors should introduce works relating on acethylcolinesterase inhibitors, for instance that of Adebambo et al. Int J Alzheimers Dis. 2024 Feb 8;2024:2988685, or that of Han et al, In Silico Screening of Small Molecule Inhibitors for Amyloid-β Aggregation. J Chem Inf Model. 2025 Jun 23;65(12):6238-6248. I also suggest to report, for instance, the papers of Barresi et al, Molecules 2024 May 3;29(9):2127 and Ciaglia et al Antioxidants 2024, 13(12), 1585. Indeed, in this framework, also gut microbiota deriving indoles or other compounds as SCFA can be part of molecules showing anti-AD or pro-AD properties, thus interesting for therapeutic developments. Important: I apologize that I did not notice this issue previously: Monoclonal antibodies therapeutic approaches should not be contained in this small molecule section! These are in fact biological macromolecules, proteins, with high molecular weight. Therefore, at Line 935, authors have to split and create a 7.4 subsection entitled: "Protein-peptide drug therapies". In this section, the authors should report works on monoclonal antibodies but also other protein-based approaches. At Line 971 the authors should consider protein based strategies linked to the gut microbiota axis and immune/inflammation. For instance, authors should report those works targeting gut microbiota dysbiosis or deep studies on proteomic microbial pro or anti- AD properties as the work of Ayan et al, Int J Mol Sci. 2023 Aug 15;24(16):12819. In addition, also short peptides, should be considered for their anti aggregation power, such as β-hairpins and derivatives, as shown by Hoyer et al. Proc Natl Acad Sci U S A. 2008 Mar 28;105(13):5099–5104. Then, paragraph 7.5 should be "Dietary intervention" and so on. At the end of Paragraph 7, maybe the authors should report a Table resuming the main works reported at each sub-paragraph:,for instance, a column showing the different therapeutic tools , probiotics, FMT, Small molecules, Protein-peptide drug therapies, dietary intervention , then a corresponding other column on the effects observed, and another one reporting the corresponding bibliographic references cited in the text of the manuscript.
Line 1103, page 23. Ever more targeted investigations are focusing on new potent and effective bioactive molecules, also based on microbiology and biotechnology of gut microbiota. Proteins, peptides and small molecules under screening for their action against neurodegenerative patterns or, conversely favoring them, both in relation to dysbiosis, represent all a rich source for screening target compounds . The intricate relationships between gut microbiota, brain and immune-inflammatory responses are also on the loop for new therapeutic strategies. Among these, molecular pathways and networks involving miRNAs, gut microbiota diversity and gene expression are acquiring relevance, as reported by Ayyanar and Vijayan, GeroScience. 2024 Nov 19;47(1):339–385. These investigations can have particular relevance on prevention from AD, as well as in the support to interventions more appropriate towards AD clinical heterogeneity and possible occurring different symptom presentations.
Comments on the Quality of English LanguageEnglish language requires, in some points , additional revision.
Round 4
Reviewer 2 Report
Comments and Suggestions for Authors
The new version of the manuscript ijms-3723397 entitled: "The Interplay of Inflammation and Gut-Microbiota Dysbiosis in Alzheimer's Disease: Mechanisms and Therapeutic Potential", written by Samat and coauthors has been provided. In this form, the paper has been considerably improved and authors have replied to all raised criticisms and issues. The manuscript is now complete and its aims are respected, providing directions to pursue for neuroscience and clinical research in this field.
Some minor comments:
Abstract: lines 34-35: "A recent study.." - Given the content of this review, a single recent study in the abstract makes little sense. The authors should instead specify: "Recent studies..." with the subject plural;
page 5, line 169: authors should check the world "synucleinopathyv"..;
page 9, line 334, figure legend: "The overproduction of microglia leads to the generation of ROS..." Overactivation sounds better in this sentence;
page 31, line 1227: If authors agree, maybe this sentence can be more relevant if completed as that: "The integration of advanced multi-omic approaches, alongside the consideration of individual and population-level differences, significantly enhances the potential to mitigate cognitive decline and improve outcomes for patients with AD, enabling therapeutic approaches diversified in respect to clinical features and progressive conditions. "
Author Response
The new version of the manuscript ijms-3723397 entitled: "The Interplay of Inflammation and Gut-Microbiota Dysbiosis in Alzheimer's Disease: Mechanisms and Therapeutic Potential", written by Samat and coauthors has been provided. In this form, the paper has been considerably improved and authors have replied to all raised criticisms and issues. The manuscript is now complete and its aims are respected, providing directions to pursue for neuroscience and clinical research in this field.
Abstract: lines 34-35: "A recent study.." - Given the content of this review, a single recent study in the abstract makes little sense. The authors should instead specify: "Recent studies..." with the subject plural.
Response: Thank you for the suggestion. The sentence has been revised following the reviewer’s recommendation.
page 5, line 169: authors should check the world "synucleinopathyv"..
Response: Thank you for the suggestion. The sentence has been revised following the reviewer’s recommendation.
page 9, line 334, figure legend: "The overproduction of microglia leads to the generation of ROS..." Overactivation sounds better in this sentence.
Response: Thank you for the suggestion. The sentence has been revised following the reviewer’s recommendation.
page 31, line 1227: If authors agree, maybe this sentence can be more relevant if completed as that: "The integration of advanced multi-omic approaches, alongside the consideration of individual and population-level differences, significantly enhances the potential to mitigate cognitive decline and improve outcomes for patients with AD, enabling therapeutic approaches diversified in respect to clinical features and progressive conditions.
Response: Thank you for the suggestion. The sentence has been revised following the reviewer’s recommendation.